# FLOW MATCHING ON GENERAL GEOMETRIES

**Ricky T. Q. Chen**
FAIR, Meta
rtqichen@meta.com

**Yaron Lipman**
FAIR, Meta and Weizmann Institute of Science
ylipman@meta.com

## ABSTRACT

We propose Riemannian Flow Matching (RFM), a simple yet powerful framework for training continuous normalizing flows on manifolds. Existing methods for generative modeling on manifolds either require expensive simulation, are inherently unable to scale to high dimensions, or use approximations for limiting quantities that result in biased training objectives. Riemannian Flow Matching bypasses these limitations and offers several advantages over previous approaches: it is simulation-free on simple geometries, does not require divergence computation, and computes its target vector field in closed-form. The key ingredient behind RFM is the construction of a relatively simple premetric for defining target vector fields, which encompasses the existing Euclidean case. To extend to general geometries, we rely on the use of spectral decompositions to efficiently compute premetrics on the fly. Our method achieves state-of-the-art performance on many real-world non-Euclidean datasets, and we demonstrate tractable training on general geometries, including triangular meshes with highly non-trivial curvature and boundaries.

## 1 INTRODUCTION

While generative models have recently made great advances in fitting data distributions in Euclidean spaces, there are still challenges in dealing with data residing in non-Euclidean spaces, specifically on general manifolds. These challenges include scalability to high dimensions (*e.g.*, (Rozen et al., 2021)), the requirement for simulation or iterative sampling during training even for simple geometries like hyperspheres (*e.g.*, (Mathieu & Nickel, 2020; De Bortoli et al., 2022)), and difficulties in constructing simple and scalable training objectives.

In this work, we introduce Riemannian Flow Matching (RFM), a simple yet powerful methodology for learning continuous normalizing flows (CNFs; (Chen et al., 2018)) on general Riemannian manifolds $\mathcal{M}$. RFM builds upon the Flow Matching framework (Lipman et al., 2023; Albergo & Vanden-Eijnden, 2023; Liu et al., 2023) and learns a CNF by regressing an implicitly defined target vector field $u_t(x)$ that pushes a base distribution $p$ towards a target distribution $q$ defined by the training examples. To address the intractability of $u_t(x)$, we employ a similar approach to Conditional Flow Matching (Lipman et al., 2023), where we regress onto conditional vector fields $u_t(x|x_1)$ that push $p$ towards individual training examples $x_1$.

A key observation underlying our Riemannian generalization is that the conditional vector field necessary for training the CNF can be explicitly expressed in terms of a "premetric" $d(x, y)$, which distinguishes pairs of points $x$ and $y$ on the manifold. A natural choice for such a premetric is the geodesic distance function, which coincides with the straight trajectories previously used in Euclidean space by prior approaches.

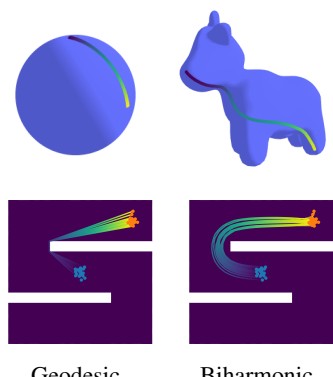

Geodesic          Biharmonic

Figure 1: Our approach makes use of user-specified *premetrics* on general manifolds to define flows. On select simple manifolds, the geodesic can be computed exactly and leads to a *simulation-free* algorithm. On general manifolds where the geodesic is not only computationally expensive but can lead to degeneracy (*e.g.*, along boundaries), we propose the use of *spectral* distances (*e.g.*, biharmonic), which can be computed efficiently contingent on a one-time processing cost.

On simple geometries, where geodesics are known in closed form (*e.g.*, Euclidean space, hypersphere, hyperbolic space, torus, or any of their product spaces), Riemannian Flow Matching remains completely simulation-free. Even on general geometries, it only requires forward simulation of a relatively simple ordinary differential equation (ODE), without differentiation through the solver, stochastic iterative sampling, or divergence estimation.

On all types of geometries, Riemannian Flow Matching offers several advantages over recently proposed Riemannian diffusion models (De Bortoli et al., 2022; Huang et al., 2022). These advantages include avoiding iterative simulation of a noising process during training for geometries with analytic geodesic formulas; not relying on approximations of score functions or divergences of the parameteric vector field; and not needing to solve stochastic differential equations (SDE) on manifolds, which is generally more challenging to approximate than ODE solutions (Kloeden et al., 2002; Hairer et al., 2006; Hairer, 2011). Table 1 summarizes the key differences with relevant prior methods, which we expand on further in Section 4 (Related Work).

|  | Simulation-free on simple geo. | Closed-form target vector field | Does not require divergence |
|---|---|---|---|
| Ben-Hamu et al. (2022) | ✓ | - | ✗ |
| De Bortoli et al. (2022) (DSM) | ✗ | ✗ | ✓ |
| De Bortoli et al. (2022) (ISM) | ✗ | - | ✗ |
| Huang et al. (2022) | ✗ | - | ✗ |
| Riemannian FM (*Ours*) | ✓ | ✓ | ✓ |

Table 1: Comparison of closely related methods for training continuous-time generative models on Riemannian manifolds. Additionally, we are the only among these works to consider and tackle general geometries.

Empirically, we find that Riemannian Flow Matching achieves state-of-the-art performance on manifold datasets across various settings, being on par or outperforming competitive baselines. We also demonstrate that our approach scales to higher dimensions without sacrificing performance, thanks to our scalable closed-form training objective. Moreover, we present the first successful training of continuous-time deep generative models on non-trivial geometries, including those imposed by discrete triangular meshes and manifolds with non-trivial boundaries that represent challenging constraints on maze-shaped manifolds.

## 2 PRELIMINARIES

**Riemannian manifolds.** This paper considers complete connected, smooth Riemannian manifolds $\mathcal{M}$ with metric $g$ as basic domain over which the generative model is learned. Tangent space to $\mathcal{M}$ at $x \in \mathcal{M}$ is denoted $T_x\mathcal{M}$, and $g$ defines an inner product over $T_x\mathcal{M}$ denoted $\langle u, v \rangle_g$, $u, v \in T_x\mathcal{M}$. $T\mathcal{M} = \cup_{x \in \mathcal{M}} \{x\} \times T_x\mathcal{M}$ is the tangent bundle that collects all the tangent planes of the manifold. $\mathcal{U} = \{u_t\}$ denotes the space of time dependent smooth vector fields (VFs) $u_t : [0, 1] \times \mathcal{M} \to T\mathcal{M}$, where $u_t(x) \in T_x\mathcal{M}$ for all $x \in \mathcal{M}$; $\mathrm{div}_g(u_t)$ is the Riemannian divergence w.r.t. the spatial ($x$) argument. We will denote by $d\mathrm{vol}_x$ the volume element over $\mathcal{M}$, and integration of a function $f : \mathcal{M} \to \mathbb{R}$ over $\mathcal{M}$ is denoted $\int f(x) d\mathrm{vol}_x$. For readers who are looking for a more comprehensive background on Riemannian manifolds, we recommend Gallot et al. (1990).

**Probability paths and flows on manifolds.** Probability densities over $\mathcal{M}$ are continuous non-negative functions $p : \mathcal{M} \to \mathbb{R}_+$ such that $\int p(x) d\mathrm{vol}_x = 1$. The space of probability densities over $\mathcal{M}$ is marked $\mathcal{P}$. A *probability path* $p_t$ is a curve in probability space $p_t : [0, 1] \to \mathcal{P}$; such paths will be used as supervision signal for training our generative models. A *flow* is a diffeomorphism $\Psi : \mathcal{M} \to \mathcal{M}$ defined by integrating instantaneous deformations represented by a time-dependent vector field $u_t \in \mathcal{U}$. Specifically, a time-dependent flow, $\psi_t : \mathcal{M} \to \mathcal{M}$, is defined by solving the following ordinary differential equation (ODE) on $\mathcal{M}$ over $t \in [0, 1]$,

$$\frac{d}{dt}\psi_t(x) = u_t(\psi_t(x)), \qquad \psi_0(x) = x, \tag{1}$$

and the final diffeomorphism is defined by setting $\Psi(x) = \psi_1(x)$. Given a probability density path $p_t$, it is said to be *generated* by $u_t$ from $p$ if $\psi_t$ pushes $p_0 = p$ to $p_t$ for all $t \in [0, 1]$. More formally,

$$\log p_t(x) = \log([\psi_t]_\sharp p)(x) = \log p(\psi_t^{-1}(x)) - \int_0^t \mathrm{div}_g(u_t)(x_s)\mathrm{d}s \tag{2}$$

where the $\sharp$ symbol denotes the standard push-forward operation and $x_s = \psi_s(\psi_t^{-1}(x))$. This formula can be derived from the Riemannian version of the instantaneous change of variables Formula (see

equation 22 in (Ben-Hamu et al., 2022)). Previously, Chen et al. (2018) suggested modeling the flow $\psi_t$ implicitly by considering parameterizing the vector field $u_t$. This results in a deep generative model of the flow $\psi_t$, called a *Continuous Normalizing Flow* (CNF) which models a probability path $p_t$ through a continuous-time deformation of a base distribution $p$. A number of works have formulated manifold variants (Mathieu & Nickel, 2020; Lou et al., 2020; Falorsi, 2020) that require simulation in order to enable training, while some simulation-free variants (Rozen et al., 2021; Ben-Hamu et al., 2022) scale poorly to high dimensions and do not readily adapt to general geometries.

## 3    METHOD

We aim to train a generative model that lies on a complete, connected smooth Riemannian manifold $\mathcal{M}$ endowed with a metric $g$. Concretely, we are given a set of training samples $x_1 \in \mathcal{M}$ from some unknown data distribution $q(x_1)$, $q \in \mathcal{P}$. Our goal is to learn a parametric map $\Phi : \mathcal{M} \to \mathcal{M}$ that pushes a simple base distribution $p \in \mathcal{P}$ to $q$.

### 3.1    FLOW MATCHING ON MANIFOLDS

Flow Matching Lipman et al. (2023) is a method to train Continuous Normalizing Flow (CNF) on Euclidean space that sidesteps likelihood computation during training and scales extremely well, similar to diffusion models (Ho et al., 2020; Song et al., 2020b), while allowing the design of more general noise processes which enables this work. We provide a brief summary and make the necessary adaptation to formulate Flow Matching on Riemannian manifolds. Derivations of the manifold case with full technical details are in Appendix A.

**Riemannian Flow Matching.**    Flow Matching (FM) trains a CNF by fitting a vector field $v \in \mathcal{U}$, *i.e.*, $v_t(x) \in T_x\mathcal{M}$, with parameters $\theta \in \mathbb{R}^p$, to an *a priori* defined target vector field $u \in \mathcal{U}$ that is known to generate a probability density path $p_t \in \mathcal{P}$ over $\mathcal{M}$ satisfying $p_0 = p$ and $p_1 = q$. On a manifold endowed with a Riemannian metric $g$, the Flow Matching objective compares the tangent vectors $v_t(x), u_t(x) \in T_x\mathcal{M}$ using the Riemannian metric $g$ at that tangent space:

$$\mathcal{L}_{\text{RFM}}(\theta) = \mathbb{E}_{t,p_t(x)} \left\| v_t(x) - u_t(x) \right\|_g^2 \tag{3}$$

where $t \sim \mathcal{U}[0, 1]$, the uniform distribution over $[0, 1]$.

**Probability path construction.**    Riemannian Flow Matching therefore requires coming up with a probability density path $p_t \in \mathcal{P}$, $t \in [0, 1]$ that satisfies the boundary conditions

$$p_0 = p, \qquad p_1 = q \tag{4}$$

and a corresponding vector field (VF) $u_t(x)$ which generates $p_t(x)$ from $p$ in the sense of equation 2. One way to construct such a pair is to create per-sample *conditional probability paths* $p_t(x|x_1)$ satisfying

$$p_0(x|x_1) = p(x), \quad p_1(x|x_1) \approx \delta_{x_1}(x), \tag{5}$$

where $\delta_{x_1}(x)$ is the Dirac distribution over $\mathcal{M}$ centered at $x_1$. One can then define $p_t(x)$ as the marginalization of these conditional probability paths over $q(x_1)$.

$$p_t(x) = \int_{\mathcal{M}} p_t(x|x_1)q(x_1)d\text{vol}_{x_1}, \tag{6}$$

which satisfies equation 4 by construction. It was then proposed by Lipman et al. (2023)—which we verify for the manifold setting—to define $u_t(x)$ as the "marginalization" of conditional vector fields $u_t(x|x_1)$ that *generates* $p_t(x|x_1)$ (in the sense detailed in Section 2),

$$u_t(x) = \int_{\mathcal{M}} u_t(x|x_1) \frac{p_t(x|x_1)q(x_1)}{p_t(x)} d\text{vol}_{x_1}, \tag{7}$$

which provably generates $p_t(x)$. However, directly plugging $u_t(x)$ into equation 3 is intractable as computing $u_t(x)$ is intractable.

**Riemmanian Conditional Flow Matching.**    A key insight from Lipman et al. (2023) is that when the targets $p_t$ and $u_t$ are defined as in equations 6 and 7, the FM objective is equivalent to the following Conditional Flow Matching objective,

$$\mathcal{L}_{\text{RCFM}}(\theta) = \mathbb{E}_{t,q(x_1),p_t(x|x_1)} \left\| v_t(x) - u_t(x|x_1) \right\|_g^2 \tag{8}$$

as long as $u_t(x|x_1)$ is a vector field that generates $p_t(x|x_1)$ from $p$.

To simplify this loss, consider the *conditional flow*, which we denote via the shorthand,

$$x_t = \psi_t(x_0|x_1), \qquad (9)$$

defined as the solution to the ODE in equation 1 with the VF $u_t(x|x_1)$ and the initial condition $\psi_0(x_0|x_1) = x_0$. Furthermore, since sampling from $p_t(x|x_1)$ can be done with $\psi_t(x_0|x_1)$, where $x_0 \sim p(x_0)$, we can reparametrize equation 8 as

$$\mathcal{L}_{\text{RCFM}}(\theta) = \mathbb{E}_{t, q(x_1), p(x_0)} \left\| v_t(x_t) - \dot{x}_t \right\|_g^2 \quad (10)$$

where $\dot{x}_t = {}^{d}/{}_{dt}\, x_t = u_t(x_t|x_1)$.

Riemannian Conditional Flow Matching (RCFM) has three requirements: a parametric vector field $v_t$ that outputs vectors on the tangent planes, the

---

**Algorithm 1** Riemannian CFM

**Require:** base $p$, target $q$, scheduler $\kappa$
  Initialize parameters $\theta$ of $v_t$
  **while** not converged **do**
    sample time $t \sim \mathcal{U}(0,1)$
    sample training example $x_1 \sim q$
    sample noise $x_0 \sim p$
    **if** simple geometry **then**
      $x_t = \exp_{x_1}(\kappa(t)\log_{x_1}(x_0))$
    **else if** general geometry **then**
      $x_t = \texttt{solve\_ODE}([0, t], x_0, u_t(x|x_1))$
    **end if**
    $\ell(\theta) = \|v_t(x_t; \theta) - \dot{x}_t\|_g^2$
    $\theta = \texttt{optimizer\_step}(\ell(\theta))$
  **end while**

---

use of the appropriate Riemannian metric $\|\cdot\|_g$, and the design of a (computationally tractable) conditional flow $\psi_t(x|x_1)$ whose probability path satisfies the boundaries conditions in equation 5. We discuss this last point in the next section. Generally, compared to existing methods for training generative models on manifolds, RCFM is both simple and highly scalable; the training procedure is summarized in Algorithm 1 and a detailed comparison can be found in Appendix D.

## 3.2 CONSTRUCTING CONDITIONAL FLOWS THROUGH PREMETRICS

We discuss the construction of conditional flows $\psi_t(x|x_1)$ on $\mathcal{M}$ that concentrate all mass at $x_1$ at time $t = 1$; Figure 2 provides an illustration. This ensures that equation 5 will hold (regardless of the choice of $p$) since all points are mapped to $x_1$ at time $t = 1$, namely

$$\psi_1(x|x_1) = x_1, \text{ for all } x \in \mathcal{M}. \qquad (11)$$

On general manifolds, directly constructing $\psi_t$ that satisfies equation 11 can be overly cumbersome. Alternatively, we propose an approach based on designing a *premetric* instead, which has simple properties that, when satisfied, characterize conditional flows which satisfy equation 11. Specifically, we define a premetric as d $: \mathcal{M} \times \mathcal{M} \to \mathbb{R}$ satisfying:

1. *Non-negative:* $\mathrm{d}(x, y) \geq 0$ for all $x, y \in \mathcal{M}$.
2. *Positive:* $\mathrm{d}(x, y) = 0$ iff $x = y$.
3. *Non-degenerate:* $\nabla \mathrm{d}(x, y) \neq 0$ iff $x \neq y$.

We use as convention $\nabla \mathrm{d}(x, y) = \nabla_x \mathrm{d}(x, y)$. Such a premetric denotes the closeness of a point $x$ to $x_1$, and we aim to design a conditional flow $\psi_t(x|x_1)$ that monotonically decreases this

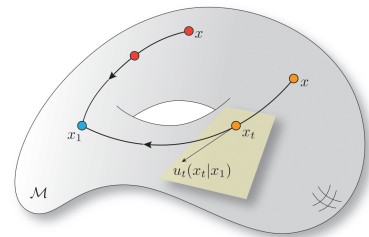

Figure 2: The conditional vector field $u_t(x|x_1)$ defined in equation 13 transports all points $x \neq x_1$ to $x_1$ at exactly $t = 1$.

premetric. That is, given a monotonically decreasing differentiable function $\kappa(t)$ satisfying $\kappa(0) = 1$ and $\kappa(1) = 0$, we want to find a $\psi_t$ that decreases $\mathrm{d}(\cdot, x_1)$ according to

$$\mathrm{d}(\psi_t(x_0|x_1), x_1) = \kappa(t)\mathrm{d}(x_0, x_1), \qquad (12)$$

Here $\kappa(t)$ acts as a scheduler that determines the rate at which $\mathrm{d}(\cdot|x_1)$ decreases. Note that at $t = 1$, we necessarily satisfy equation 11 since $x_1$ is the unique solution to $\mathrm{d}(\cdot|x_1) = 0$ due to the "positive" property of the premetric. Our next theorem shows that $\psi_t(x|x_1)$ satisfying equation 12 results in the following vector field,

$$u_t(x|x_1) = \frac{d \log \kappa(t)}{dt} \mathrm{d}(x, x_1) \frac{\nabla \mathrm{d}(x, x_1)}{\|\nabla \mathrm{d}(x, x_1)\|_g^2}, \qquad (13)$$

The "non-degenerate" property guarantees this conditional vector field is defined everywhere $x \neq x_1$.

**Theorem 3.1.** *The flow $\psi_t(x|x_1)$ defined by the vector field $u_t(x|x_1)$ in equation 13 satisfies equation 12, and therefore also equation 11. Conversely, out of all conditional vector fields that satisfy equation 12, this $u_t(x|x_1)$ is the minimal norm solution.*

A more concise statement and full proof of this result can be found in Appendix B. Here we provide proof for the first part: Consider the scalar function $a(t) = \mathrm{d}(x_t, x_1)$, where $x_t = \psi_t(x|x_1)$ is the flow defined with the VF in equation 13. Differentiation w.r.t. time gives

$$\frac{d}{dt}a(t) = \langle \nabla \mathrm{d}(x_t, x_1), \dot{x}_t \rangle_g = \langle \nabla \mathrm{d}(x_t, x_1), u(x_t|x_1) \rangle_g = \frac{d \log \kappa(t)}{dt} a(t),$$

The solution of this ODE is $a(t) = \kappa(t)\mathrm{d}(x, x_1)$, which can be verified through substitution, and hence proves $\mathrm{d}(x_t, x_1) = \kappa(t)\mathrm{d}(x, x_1)$. Intuitively, $u_t(x|x_1)$ is the minimal norm solution since it does not contain orthogonal directions that do not decrease the premetric.

A simple choice we make in this paper for the scheduler is $\kappa(t) = 1 - t$, resulting in a conditional flow that linearly decreases the premetric between $x_t$ and $x_1$. Using this, we arrive at a more explicit form of the RCFM objective,

$$\mathcal{L}_{\text{RCFM}}(\theta) = \mathbb{E}_{t,q(x_1),p(x_0)} \left\| v_t(x_t) + \mathrm{d}(x_0, x_1) \frac{\nabla \mathrm{d}(x_t, x_1)}{\|\nabla \mathrm{d}(x_t, x_1)\|_g^2} \right\|_g^2. \tag{14}$$

For general manifolds $\mathcal{M}$ and premetrics d, training with Riemmanian CFM will require simulation in order to solve for $x_t$, though it does not need to differentiate through $x_t$. However, on simple geometries RCFM can become completely simulation-free by choosing the premetric to be the geodesic distance, as we discuss next.

**Geodesic distance.** A natural choice for the premetric $\mathrm{d}(x, y)$ over a Riemannian manifold $\mathcal{M}$ is the geodesic distance $\mathrm{d}_g(x, y)$. Firstly, we note that when using geodesic distance as our choice of premetric, the flow $\psi_t(x_0|x_1)$—since it is the minimal norm solution—is equivalent to the geodesic path, *i.e.*, shortest path, connecting $x_0$ and $x_1$.

**Proposition 3.2.** *Consider a complete, connected smooth Riemannian manifold $(\mathcal{M}, g)$ with geodesic distance $d_g(x, y)$. In case $d(x, y) = d_g(x, y)$ then $x_t = \psi_t(x_0|x_1)$ defined by the conditional VF in equation 13 with the scheduler $\kappa(t) = 1 - t$ is a constant speed geodesic connecting $x_0$ to $x_1$.*

This makes it easy to compute $x_t$ on *simple* manifolds, *which we define as manifolds with closed-form geodesics*, *e.g.*, Euclidean space, the hypersphere, the hyperbolic space, the high-dimensional torus, and some matrix Lie Groups. In particular, the geodesic connecting $x_0$ and $x_1$ can be expressed in terms of the exponential and logarithm maps,

$$x_t = \exp_{x_1}(\kappa(t) \log_{x_1}(x_0)), \quad t \in [0, 1]. \tag{15}$$

This formula can simply be plugged into equation 10, resulting in a highly scalable training objective. A list of simple manifolds that we consider can be found in Table 5.

**Euclidean geometry.** With Euclidean geometry $\mathcal{M} = \mathbb{R}^n$, and with standard Euclidean norm $\mathrm{d}(x, y) = \|x - y\|_2$, the conditional VF (equation 13) with scheduler $\kappa(t) = 1 - t$ reduces to the VF used by Lipman et al. (2023), $u_t(x|x_1) = \frac{x_1 - x}{1-t}$, and the RCFM objective takes the form

$$\mathcal{L}_{\text{RCFM}}(\theta) = \mathbb{E}_{t,q(x_1),p(x_0)} \|v_t(x_t) + x_0 - x_1\|_2^2,$$

which coincides with the Euclidean case of Flow Matching presented in prior works (Lipman et al., 2023; Liu et al., 2023).

## 3.3 SPECTRAL DISTANCES ON GENERAL GEOMETRIES

Geodesics can be difficult to compute efficiently for general geometries, especially since it needs to be computed for any possible pair of points. Hence, we propose using premetrics that can be computed quickly for any pair of points on $\mathcal{M}$ contingent on a one-time upfront cost. In particular, for general Riemannian manifolds, we consider the use of approximate spectral distances as an alternative to the geodesic distance. Spectral distances actually offer some benefits over the geodesic distance such as

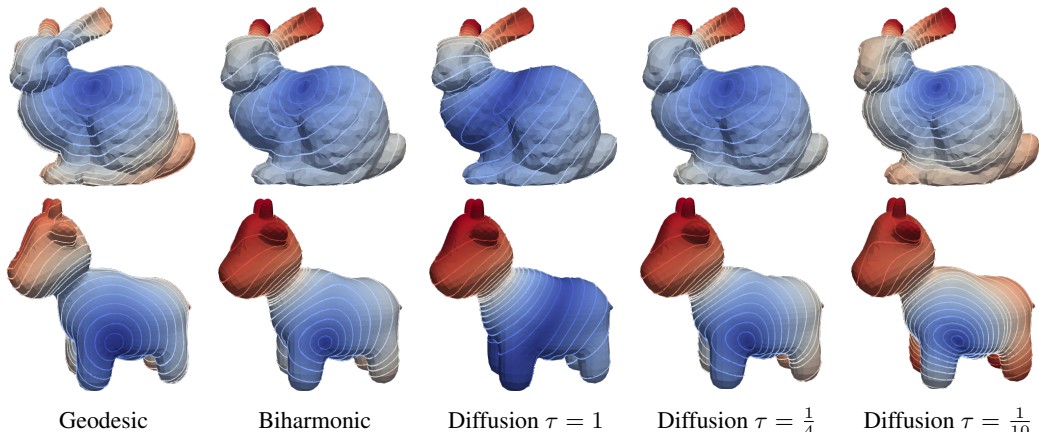

| Geodesic | Biharmonic | Diffusion $\tau = 1$ | Diffusion $\tau = \frac{1}{4}$ | Diffusion $\tau = \frac{1}{10}$ |

Figure 3: Contour plots of geodesic and spectral distances (to a source point) on general manifolds. Geodesics are expensive to compute online and are globally non-smooth. The biharmonic distance behaves smoothly while the diffusion distance requires careful tuning of the hyperparameter $\tau$.

robustness to topological noise, smoothness, and are globally geometry-aware (Lipman et al., 2010). Note however, that spectral distances do not define minimizing (geodesic) paths, and will require simulation of $u_t(x|x_1)$ in order to compute conditional flows $x_t$.

Let $\varphi_i : \mathcal{M} \to \mathbb{R}$ be the eigenfunctions of the Laplace-Beltrami operator $\Delta_g$ over $\mathcal{M}$ with corresponding eigenvalues $\lambda_i$, *i.e.*, they satisfy $\Delta_g \varphi_i = \lambda_i \varphi_i$, for $i = 1, 2, \ldots$, then spectral distances are of the form

$$\mathrm{d}_w(x, y)^2 = \sum_{i=1}^{\infty} w(\lambda_i) \left( \varphi_i(x) - \varphi_i(y) \right)^2, \tag{16}$$

where $w : \mathbb{R} \to \mathbb{R}_+$ is some monotonically decreasing weighting function. Popular instances of spectral distances include:

1. *Diffusion Distance* Coifman & Lafon (2006): $w(\lambda) = \exp(-2\tau\lambda)$, with a parameter $\tau$.
2. *Biharmonic Distance* Lipman et al. (2010): $w(\lambda) = \lambda^{-2}$.

In practice, we truncate the infinite series in equation 16 to the smallest $k$ eigenvalues. These $k$ eigenfunctions can be numerically solved as a **one-time preprocessing cost** prior to training. Furthermore, we note that using an approximation of the spectral distance with **finite** $k$ **is sufficient** for satisfying the properties of the premetric, leading to no bias in the training procedure. Lastly, we consider **manifolds with boundaries** and show that solving eigenfunctions using the natural, or Neumann, boundary conditions ensures that the resulting $u_t(x|x_1)$ does not leave the interior of the manifold. Detailed discussions on these points above can be found in Appendix G. Figure 3 visualizes contour plots of these spectral distances for manifolds with non-trivial curvatures.

## 4 RELATED WORK

**Deep generative models on Riemannian manifolds.** Some initial works suggested constructing normalizing flows that map between manifolds and Euclidean spaces of the same intrinsic dimension (Gemici et al., 2016; Rezende et al., 2020; Bose et al., 2020), often relying on the tangent space at some pre-specified origin. However, this approach is problematic when the manifold is not homeomorphic to Euclidean space, resulting in both theoretical and numerical issues. On the other hand, continuous-time models such as continuous normalizing flows bypass such topogical constraints and flow directly on the manifold itself. To this end, a number of works have formulated continuous normalizing flows on simple manifolds (Mathieu & Nickel, 2020; Lou et al., 2020; Falorsi, 2020), but these rely on maximum likelihood for training, a costly simulation-based procedure. More recently, simulation-free training methods for continuous normalizing flows on manifolds have been proposed (Rozen et al., 2021; Ben-Hamu et al., 2022); however, these scale poorly to high dimensions and do not adapt to general geometries.

Table 2: Test NLL on Earth and climate science datasets. Standard deviation estimated over 5 runs.

| | Volcano | Earthquake | Flood | Fire |
|---|---|---|---|---|
| Dataset size (train + val + test) | 827 | 6120 | 4875 | 12809 |
| *CNF-based* | | | | |
| Riemannian CNF (Mathieu & Nickel, 2020) | -6.05$_{\pm 0.61}$ | 0.14$_{\pm 0.23}$ | 1.11$_{\pm 0.19}$ | -0.80$_{\pm 0.54}$ |
| Moser Flow (Rozen et al., 2021) | -4.21$_{\pm 0.17}$ | -0.16$_{\pm 0.06}$ | 0.57$_{\pm 0.10}$ | -1.28$_{\pm 0.05}$ |
| CNF Matching (Ben-Hamu et al., 2022) | -2.38$_{\pm 0.17}$ | -0.38$_{\pm 0.01}$ | **0.25**$_{\pm 0.02}$ | -1.40$_{\pm 0.02}$ |
| Riemannian Score-Based (De Bortoli et al., 2022) | -4.92$_{\pm 0.25}$ | -0.19$_{\pm 0.07}$ | 0.48$_{\pm 0.17}$ | -1.33$_{\pm 0.06}$ |
| *ELBO-based* | | | | |
| Riemannian Diffusion Model (Huang et al., 2022) | -6.61$_{\pm 0.96}$ | **-0.40**$_{\pm 0.05}$ | 0.43$_{\pm 0.07}$ | -1.38$_{\pm 0.05}$ |
| *Ours* | | | | |
| Riemannian Flow Matching ᵂ/ Geodesic | **-7.93**$_{\pm 1.67}$ | -0.28$_{\pm 0.08}$ | 0.42$_{\pm 0.05}$ | **-1.86**$_{\pm 0.11}$ |

**Riemannian diffusion models.**   With the influx of diffusion models that allow efficient simulation-free training on Euclidean space (Ho et al., 2020; Song et al., 2020b), multiple works have attempted to adopt diffusion models to manifolds (Mathieu & Nickel, 2020; Huang et al., 2022). However, due to the reliance on stochastic differential equations (SDE) and denoising score matching (Vincent, 2011), these approaches necessitate in-training simulation and approximations when applied to non-Euclidean manifolds.

First and foremost, they lose the simulation-free sampling of $x_t \sim p_t(x|x_1)$ that is offered in the Euclidean regime; this is because the manifold analog of the Ornstein–Uhlenbeck SDE does not have closed-form solutions. Hence, diffusion-based methods have to resort to simulated random walks as a noising process even on simple manifolds (De Bortoli et al., 2022; Huang et al., 2022).

Furthermore, even on simple manifolds, the conditional score function is not known analytically, so De Bortoli et al. (2022) proposed approximating the conditional score function with either an eigenfunction expansion or Varhadan's heat-kernel approximation. These approximations lead to biased gradients in the denoising score matching framework. We find that the heat-kernel approximations can potentially be extremely biased, even with hundreds with eigenfunctions (see Figure 10). In contrast, we show in Figure 11 that our framework can satisfy all premetric requirements even with a small number of eigenfunctions for the spectral distance approximation—hence guaranteeing that the optimal model distribution is the data distribution. See detailed discussions in Appendix G.1.

A way to bypass the conditional score function is to use implicit score matching (Hyvärinen & Dayan, 2005), which Huang et al. (2022) adopts for the manifold case, but this instead requires divergence computation of the large neural nets during training. Using the Hutchinson estimator (Hutchinson, 1989; Skilling, 1989; Grathwohl et al., 2019; Song et al., 2020a) for divergence estimation results in a more scalable algorithm, but the variance of the Hutchinson estimator scales poorly with dimension (Hutchinson, 1989) and is further exacerbated on non-Euclidean manifolds (Mathieu & Nickel, 2020).

Finally, the use of SDEs as a noising process requires carefully constructing suitable reverse-time processes that approximate either just the probability path (Anderson, 1982) or the actual sample trajectories (Li et al., 2020), whereas ODE solutions are generally well-defined in both forward and reverse directions (Murray & Miller, 2013).

In contrast to these methods, Riemannian Flow Matching is simulation-free on simple geometries, has exact conditional vector fields, and does not require divergence computation during training. These properties are summarized in Table 1, and a detailed comparison of algorithmic differences to diffusion-based approaches is presented in Appendix D. Lastly, for general Riemannian manifolds we show that the design of a relatively simple *premetric* is sufficient, allowing the use of general distance functions that don't satisfy all axioms of a metric—such as approximate spectral distances with finite truncation—going beyond what is currently possible with existing Riemannian diffusion methods.

**Euclidean Flow Matching.**   Riemannian Flow Matching is built on top of recent simulation-free methods that work with ODEs instead of SDEs, regressing directly onto generating vector fields instead of score functions (Lipman et al., 2023; Albergo & Vanden-Eijnden, 2023; Liu et al., 2023; Neklyudov et al., 2022), resulting in an arguably simpler approach to continuous-time generative modeling without the intricacies of dealing with stochastic differential equations. In particular, Lipman et al. (2023) shows that this approach encompasses and broadens the probability paths used

Table 3: Test NLL on protein datasets. Standard deviation estimated over 5 runs.

| | General (2D) | Glycine (2D) | Proline (2D) | Pre-Pro (2D) | RNA (7D) |
|---|---|---|---|---|---|
| Dataset size (train + val + test) | 138208 | 13283 | 7634 | 6910 | 9478 |
| Mixture of Power Spherical (Huang et al., 2022) | $1.15_{\pm0.002}$ | $2.08_{\pm0.009}$ | $0.27_{\pm0.008}$ | $1.34_{\pm0.019}$ | $4.08_{\pm0.368}$ |
| Riemannian Diffusion Model (Huang et al., 2022) | $1.04_{\pm0.012}$ | $1.97_{\pm0.012}$ | $\mathbf{0.12}_{\pm\mathbf{0.011}}$ | $1.24_{\pm0.004}$ | $-3.70_{\pm0.592}$ |
| Riemannian Flow Matching w/ Geodesic | $\mathbf{1.01}_{\pm\mathbf{0.025}}$ | $\mathbf{1.90}_{\pm\mathbf{0.055}}$ | $0.15_{\pm0.027}$ | $\mathbf{1.18}_{\pm\mathbf{0.055}}$ | $\mathbf{-5.20}_{\pm\mathbf{0.067}}$ |

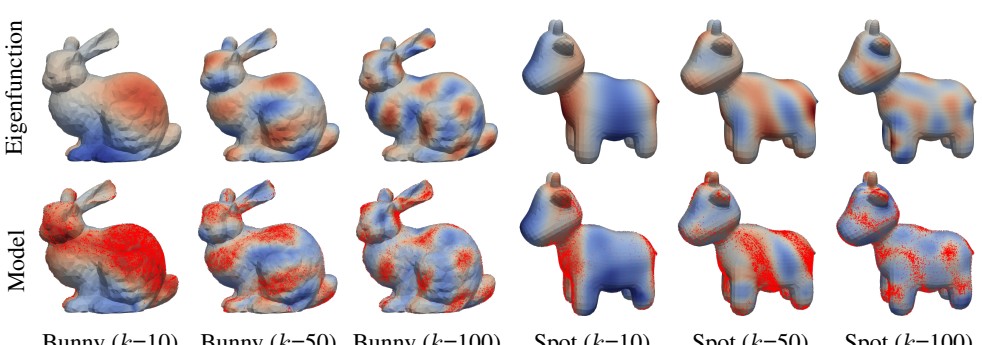

Bunny ($k$=10)  Bunny ($k$=50)  Bunny ($k$=100)  Spot ($k$=10)  Spot ($k$=50)  Spot ($k$=100)

Figure 4: Visualization of (*top*) the eigenfunctions that were used to construct target distributions, and (*bottom*) the learned density & samples from trained models with the Biharmonic distance.

by diffusion models while remaining simulation-free; Albergo & Vanden-Eijnden (2023) discusses an interpretation based on the use of interpolants—equivalent to our conditional flows $\psi_t(x|x_1)$, except we also make explicit the construction of the marginal probability path $p_t(x)$ and vector field $u_t(x)$; Liu et al. (2023) shows that repeatedly fitting to a model's own samples leads to straighter trajectories; and Neklyudov et al. (2022) formulates an implicit objective when $u_t(x)$ is a gradient field.

## 5 EXPERIMENTS

We consider data from earth and climate science, protein structures, high-dimensional tori, complicated synthetic distributions on general closed manifolds, and distributions on maze-shaped manifolds that require navigation across non-trivial boundaries. Details regarding training setup is discussed in Appendix H. Due to space constraints, additional experiments on hyperbolic manifold and a manifold over matrices can be found in Appendix J, which are endowed with nontrivial Riemannian metrics. Table 5 provides all details of the simple manifolds and their geometries. Details regarding the more complex mesh manifolds can be found in the open source code, which we release for reproducibility[1].

**Earth and climate science datasets on the sphere.** We make use of the publicly sourced datasets (NOAA, 2020a;b; Brakenridge, 2017; EOSDIS, 2020) compiled by Mathieu & Nickel (2020). These data points lie on the 2-D sphere, a simple manifold with closed form exponential and logarithm maps. We therefore stick to the geodesic distance and compute geodesics in closed form as in equation 15. Table 2 shows the results alongside prior methods. We achieve a sizable improvement over prior works on the volcano and fire datasets which have highly concentrated regions that require a high fidelity. Figure 8 shows the density of our trained models.

**Protein datasets on the torus.** We make use of the preprocessed protein (Lovell et al., 2003) and RNA (Murray et al., 2003) datasets compiled by Huang et al. (2022). These datasets represent torsion angles and can be represented on the 2D and 7D torus. We represent the data on a flat torus, which is isometric to the product of 1-D spheres used by prior works (Huang et al., 2022; De Bortoli et al., 2022) and result in densities that are directly comparable due to this isometry. Results are displayed in Table 3, and we show learned densities of the protein datasets in Figure 9. Compared to Huang et al. (2022), we see a significant gain in performance particularly on the higher dimensional 7D torus, due to the higher complexity of the dataset.

**Scaling to high dimensions.** We next consider the scalability of our method in the case of high-dimensional tori, following the exact setup in De Bortoli et al. (2022). We com-

---

[1]https://github.com/facebookresearch/riemannian-fm

pare to Moser Flow (Rozen et al., 2021), which does not scale well into high dimensions, and Riemannian Score-based (De Bortoli et al., 2022) using implicit score matching (ISM).

As shown in Table 1, this objective gets around the need to approximate conditional score functions, but it requires stochastic divergence estimation, introducing larger amounts of variance at higher dimensions. In Figure 5 we plot log-likelihood values, across these two baselines and our method with the geodesic construction. We see that our method performs steadily, with no significant drop in performance at higher dimensions since we do not have any reliance on approximations.

Figure 5: Riemannian Flow Matching scales incredibly well to higher dimensions as it is simulation-free and all quantities required for training are computed exactly on simple geometries such as tori. Log-likelihoods are in bits.

**Manifolds with non-trivial curvature.** We next experiment with general closed manifolds using spectral distances as described in Section 3.3. Specifically, we experiment on manifolds described by triangular meshes. For meshes, computing geodesic distances on-the-fly is too expensive for our use case, which requires hundreds of evaluations per training iteration. Fast approximations to the geodesic distance between two points are $\mathcal{O}(n \log n)$ (Kimmel & Sethian, 1998), while exact geodesic distances require $\mathcal{O}(n^2 \log n)$ (Surazhsky et al., 2005), where $n$ is the number of edges. On the other hand, computing spectral distances is $\mathcal{O}(k)$, where $k \ll n$, *i.e.*, it does not scale with the complexity of the manifold after the one-time preprocessing step. As our manifolds, we use the Standard Bunny (Turk & Levoy, 1994) and Spot the Cow (Crane et al., 2013). Similar to Rozen et al. (2021), we construct distributions by computing the $k$-th eigenfunction, thresholding, and then sampling proportionally to the eigenfunction. This is done on a high-resolution mesh so that the distribution is non-trivial on each triangle. Figure 4 contains visualizations of the eigenfunctions, the learned density, and samples from a trained model that transports from a uniform base distribution. We used $k=200$ eigenfunctions, which is sufficient for our method to produce high fidelity samples.

In Table 4, we report the test NLL of models trained using either the diffusion distance or the biharmonic distance. We had to carefully tune the diffusion distance hyperparameter $\tau$ while the biharmonic distance was straightforward to use out-of-the-box and it has better smoothness properties (see Figure 3).

|  | **Stanford Bunny** | | | **Spot the Cow** | | |
|---|---|---|---|---|---|---|
|  | $k=10$ | $k=50$ | $k=100$ | $k=10$ | $k=50$ | $k=100$ |
| Riemannian CFM | | | | | | |
| w/ Diffusion ($\tau=1/4$) | $1.16_{\pm 0.02}$ | $\mathbf{1.48_{\pm 0.01}}$ | $1.53_{\pm 0.01}$ | $\mathbf{0.87_{\pm 0.07}}$ | $\mathbf{0.95_{\pm 0.16}}$ | $\mathbf{1.08_{\pm 0.05}}$ |
| w/ Biharmonic | $\mathbf{1.06_{\pm 0.05}}$ | $1.55_{\pm 0.01}$ | $\mathbf{1.49_{\pm 0.01}}$ | $1.02_{\pm 0.06}$ | $1.08_{\pm 0.05}$ | $1.29_{\pm 0.05}$ |

Table 4: Test NLL on mesh datasets.

**Manifolds with boundaries.** Lastly, we experiment with manifolds that have boundaries. Specifically, we consider randomly generated mazes visualized in Figure 6. We set the base distribution to be a Gaussian in the middle of the maze, and set the target distribution to be a mixture of densities at corners of the maze. These mazes are represented using triangular meshes, and we use the biharmonic distance using $k=30$ eigenfunctions. Once trained, the model represents a single vector field that transports all mass from the source distribution to the target distribution with no crossing paths. We plot

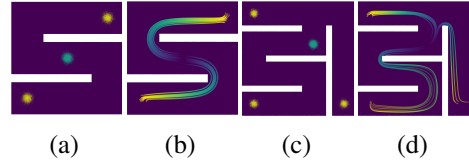

(a)      (b)      (c)      (d)

Figure 6: (*a, c*) Source (*cyan*) and target (*yellow*) distributions on a manifold with non-trivial boundaries. (*b, d*) Sample trajectories from a CNF model trained through RCFM with the Biharmonic distance.

sample trajectories in Figure 6 (b) and (d), where it can be seen that the learned vector field avoids boundaries of the manifold and successfully navigates to different modes in the target distribution.

## 6 CONCLUSION

We propose Riemannian Flow Matching as a highly-scalable approach for training continuous normalizing flows on manifolds. Our method is completely simulation-free and introduces zero approximation errors on simple geometries that have closed-form geodesics. We also introduce benchmark problems for general manifolds and showcase for the first time, tractable training on general geometries including both closed manifolds and manifolds with boundaries.

ACKNOWLEDGEMENTS

Ricky T. Q. Chen would like to thank Chin-Wei Huang for helpful discussions. Additionally, we acknowledge the Python community (Van Rossum & Drake Jr, 1995; Oliphant, 2007) for developing the core set of tools that enabled this work, including PyTorch (Paszke et al., 2019), PyTorch Lightning (Falcon & team, 2019), Hydra (Yadan, 2019), Jupyter (Kluyver et al., 2016), Matplotlib (Hunter, 2007), seaborn (Waskom et al., 2018), numpy (Oliphant, 2006; Van Der Walt et al., 2011), SciPy (Jones et al., 2014) pandas (McKinney, 2012), geopandas (Jordahl et al., 2020), torchdiffeq (Chen, 2018), libigl (Panozzo & Jacobson, 2014), and PyEVTK (Herrera, 2019).

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

## A    CONDITIONAL FLOW MATCHING ON MANIFOLDS

We provide the necessary derivations and proofs for the Conditional Flow Matching over a Riemannian manifolds; the proofs and derivations from Lipman et al. (2023) are followed "as-is", with the necessary adaptation to the Riemannian setting.

**Assumptions.** We will use notations and setup from Section 2. Let $p(\cdot|x_1) : [0,1] \to \mathcal{P}$ be a (conditional) probability path sufficiently smooth with integrable derivatives, strictly positive $p_t(x|x_1) > 0$, and $p_0(x|x_1) = p$, where $p \in \mathcal{P}$ is our source density. Let $u(\cdot|x_1) \in \mathfrak{U}$ be a (conditional) time-dependent vector field, sufficiently smooth with integrable derivatives and such that

$$\int_0^1 \int_{\mathcal{M}} \|u_t(x|x_1)\|_g \, p_t(x|x_1) d\mathrm{vol}_x dt < \infty.$$

Further assume $u_t(x|x_1)$ generates $p_t(x|x_1)$ from $p$ in the sense of equation 2, *i.e.*, if we denote by $\psi_t(x|x_1)$ the solution to the ODE (equation 1):

$$\frac{d}{dt}\psi_t(x|x_1) = u_t(\psi_t(x|x_1)|x_1) \tag{17}$$

$$\psi_0(x|x_1) = x \tag{18}$$

then

$$p_t(\cdot|x_1) = [\psi_t(\cdot|x_1)]_\# \, p. \tag{19}$$

**Proof of the marginal VF formula, equation 7.** First, the Mass Conservation Formula Theorem (see, *e.g.*, Villani (2009)) implies that $p_t(x|x_1)$ and $u_t(x|x_1)$ satisfy

$$\frac{d}{dt}p_t(x|x_1) + \mathrm{div}_g(p_t(x|x_1)u_t(x|x_1)) = 0 \tag{20}$$

where $\mathrm{div}_g$ is the Riemannian divergence with metric $g$.

Next, we differentiate the marginal $p_t(x)$ w.r.t. $t$:

$$\begin{aligned}
\frac{d}{dt}p_t(x) &= \int_{\mathcal{M}} \frac{d}{dt}p_t(x|x_1)q(x_1)d\mathrm{vol}_{x_1} \\
&= -\mathrm{div}_g\left[\int_{\mathcal{M}} u_t(x|x_1)p_t(x|x_1)q(x_1)d\mathrm{vol}_{x_1}\right] \\
&= -\mathrm{div}_g\left[p_t(x)\int_{\mathcal{M}} u_t(x|x_1)\frac{p_t(x|x_1)q(x_1)}{p_t(x)}d\mathrm{vol}_{x_1}\right] \\
&= -\mathrm{div}_g\left[p_t(x)u_t(x)\right]
\end{aligned}$$

where in the first and second equalities we changed the order of differentiation and integration, and in the second equality we used the mass conservation formula for $u_t(x|x_1)$. In the previous to last equality we multiplied and divided by $p_t(x)$. In the last equality we defined the marginal vector field $u_t$ as in equation 7.

**RCFM loss equivalent to RFM loss.** We will now show the equivalence of the RCFM loss (equation 8) and the RFM loss (equation 3). First note the losses expand as follows:

$$\mathcal{L}_{\mathrm{RFM}}(\theta) = \mathbb{E}_{t,p_t(x)} \|v_t(x) - u_t(x)\|_g^2 = \mathbb{E}_{t,p_t(x)} \|v_t(x)\|_g^2 - 2\langle v_t(x), u_t(x)\rangle_g + \|u_t(x)\|_g^2$$

$$\mathcal{L}_{\mathrm{RCFM}}(\theta) = \mathbb{E}_{\substack{t,q(x_1)\\p_t(x|x_1)}} \|v_t(x) - u_t(x|x_1)\|_g^2 = \mathbb{E}_{\substack{t,q(x_1)\\p_t(x|x_1)}} \|v_t(x)\|_g^2 - 2\langle v_t(x), u_t(x|x_1)\rangle_g + \|u_t(x|x_1)\|_g^2$$

Second, note that

$$\begin{aligned}
\mathbb{E}_{t,q(x_1),p_t(x|x_1)} \|v_t\|_g^2 &= \int_0^1 \int_{\mathcal{M}} \|v_t(x)\|_g^2 \, p_t(x|x_1)q(x_1)d\mathrm{vol}_x d\mathrm{vol}_{x_1} dt \\
&= \int_0^1 \int_{\mathcal{M}} \|v_t(x)\|_g^2 \, p_t(x)d\mathrm{vol}_x dt \\
&= \mathbb{E}_{t,p_t(x)} \|v_t\|_g^2
\end{aligned}$$

Lastly,

$$
\begin{aligned}
\mathbb{E}_{t,q(x_1),p_t(x|x_1)} \langle v_t(x), u_t(x|x_1) \rangle_g &= \int_0^1 \int_{\mathcal{M}} \int_{\mathcal{M}} \langle v_t(x), u_t(x|x_1) \rangle_g \, p_t(x|x_1) q(x_1) d\mathrm{vol}_x d\mathrm{vol}_{x_1} dt \\
&= \int_0^1 \int_{\mathcal{M}} \left\langle v_t(x), \int_{\mathcal{M}} u_t(x|x_1) p_t(x|x_1) q(x_1) d\mathrm{vol}_{x_1} \right\rangle_g d\mathrm{vol}_x \, dt \\
&= \int_0^1 \int_{\mathcal{M}} \left\langle v_t(x), \int_{\mathcal{M}} u_t(x|x_1) \frac{p_t(x|x_1) q(x_1)}{p_t(x)} d\mathrm{vol}_{x_1} \right\rangle_g p_t(x) d\mathrm{vol}_x \, dt \\
&= \int_0^1 \int_{\mathcal{M}} \langle v_t(x), u_t(x) \rangle_g \, p_t(x) d\mathrm{vol}_x \, dt \\
&= \mathbb{E}_{t,p_t(x)} \langle v_t(x), u_t(x) \rangle_g
\end{aligned}
$$

We got that $\mathcal{L}_{\mathrm{RCFM}}(\theta)$ and $\mathcal{L}_{\mathrm{RFM}}(\theta)$ differ by a constant,

$$
\text{const} = \int_0^1 \int_{\mathcal{M}} \|u_t(x)\|_g^2 \, p_t(x) d\mathrm{vol}_x \, dt - \int_0^1 \int_{\mathcal{M}} \|u_t(x|x_1)\|_g^2 \, p_t(x|x_1) q(x_1) d\mathrm{vol}_x d\mathrm{vol}_{x_1} \, dt
$$

that does not depend on $\theta$.

## B  PROOF OF THEOREM 3.1

**_Theorem 3.1._** The flow $\psi_t(x|x_1)$ defined by the vector field $u_t(x|x_1)$ in equation 13 satisfies equation 12, and therefore also equation 11. Conversely, out of all conditional flows $\tilde{\psi}_t(x|x_1)$ defined by a vector fields $\tilde{u}(x|x_1)$ that satisfy equation 12, this $u_t(x|x_1)$ is of minimal norm.

*Proof.* Let $x_t = \psi_t(x|x_1)$ be the flow defined by equation 13 in the sense of equation 1. Differentiating the time-dependent function $\mathrm{d}(x_t, x_1)$ w.r.t. time gives

$$
\frac{d}{dt} \mathrm{d}(x_t, x_1) = \langle \nabla \mathrm{d}(x_t, x_1), \dot{x}_t \rangle_g = \langle \nabla \mathrm{d}(x_t, x_1), u_t(x_t|x_1) \rangle_g = \frac{d \log \kappa(t)}{dt} \mathrm{d}(x_t, x_1) \tag{21}
$$

This shows that the function $a(t) = \mathrm{d}(x_t, x_1)$ satisfies the ODE

$$
\frac{d}{dt} a(t) = \frac{d \log \kappa(t)}{dt} a(t),
$$

with the initial condition $a(0) = \mathrm{d}(x, x_1)$. General solutions to this ODE are of the form $a(t) = c\kappa(t)$, where $c > 0$ is a constant set by the initial conditions. This can be verified by substitution. The constant $c$ is set by the initial condition,

$$
\mathrm{d}(x, x_1) = a(0) = c\kappa(0) = c.
$$

This gives $a(t) = \mathrm{d}(x, x_1)\kappa(t)$ as the solution. Due to uniqueness of ODE solutions we get that equation 12 holds.

Conversely, consider $x_t = \psi_t(x|x_1)$ satisfying equation 12. Differentiating both sides of this equation w.r.t. $t$ and then using equation 12 again we get

$$
\langle \nabla \mathrm{d}(x_t, x_1), \dot{x}_t \rangle_g = \frac{d\kappa(t)}{dt} \mathrm{d}(x, x_1) = \frac{d\kappa(t)}{dt} \frac{1}{\kappa(t)} \mathrm{d}(x_t, x_1) = \frac{d \log \kappa(t)}{dt} \mathrm{d}(x_t, x_1).
$$

If we let $u_t(x|x_1)$ denote the VF defining the diffeomorphism $\psi_t(x|x_1)$ in the sense of equation 1 then the last equation takes the form

$$
\langle \nabla \mathrm{d}(x_t, x_1), u_t(x_t|x_1) \rangle_g = \frac{d \log \kappa(t)}{dt} \mathrm{d}(x_t, x_1). \tag{22}
$$

This equation provides an under-determined linear system for $u_t(y|x_1) \in T_y\mathcal{M}$ at every point $y = x_t$ with non-zero probability, which in our case is all $y \in \mathcal{M}$ as we assume $p_t(y|x_1) > 0$ for all $y \in \mathcal{M}$. As can be seen in equation 21, $u_t(x|x_1)$ defined in equation 13 is also satisfying this equation. Further note, that since $u_t(x|x_1)$ defined in equation 13 satisfies $u_t(x|x_1) \parallel \nabla \mathrm{d}(x, x_1)$ (proportional) it is the minimal norm solution to the linear system in equation 22. $\square$

## C    PROOF OF PROPOSITION 3.2

***Proposition*** *3.2.* Consider a complete, connected smooth Riemannian manifold $(\mathcal{M}, g)$ with geodesic distance $\mathrm{d}_g(x, y)$. In case $\mathrm{d}(x, y) = \mathrm{d}_g(x, y)$ then $x_t = \psi_t(x_0|x_1)$ defined by the conditional VF in equation 13 with the scheduler $\kappa(t) = 1 - t$ is a geodesic connecting $x_0$ to $x_1$.

*Proof.* First, note that by definition $\psi_0(x_0|x_1) = x_0$, and $\psi_1(x_0|x_1) = x_1$. Second, from Proposition 6 in McCann (2001) we have that

$$\nabla_x \frac{1}{2} \mathrm{d}_g(x, y)^2 = -\log_x(y)$$

where $\log$ is the Riemannian logarithm map. From the chain rule we have

$$\nabla_x \frac{1}{2} \mathrm{d}_g(x, y)^2 = \mathrm{d}_g(x, y) \nabla_x \mathrm{d}_g(x, y)$$

Since the logarithm map satisfies $\|\log_x(y)\|_g = \mathrm{d}_g(x, y)$ we have that

$$\|\nabla \mathrm{d}_g(x, y)\|_g = 1. \tag{23}$$

Now, computing the length of the curve $x_t$ we get

$$
\begin{aligned}
\int_0^1 \|\dot{x}_t\|_g \, dt &= \int_0^1 \|u_t(x_t|x_1)\|_g \, dt \\
&= \int_0^1 \left\| -\frac{\mathrm{d}_g(x_t, x_1)}{1 - t} \frac{\nabla \mathrm{d}_g(x_t, x_1)}{\|\nabla \mathrm{d}_g(x_t, x_1)\|_g^2} \right\|_g \, dt \\
&= \mathrm{d}_g(x_0, x_1) \int_0^1 \left\| \frac{\nabla \mathrm{d}_g(x_t, x_1)}{\|\nabla \mathrm{d}_g(x_t, x_1)\|_g^2} \right\|_g \, dt \\
&= \mathrm{d}_g(x_0, x_1) \int_0^1 \, dt \\
&= \mathrm{d}_g(x_0, x_1)
\end{aligned}
$$

where in the second equality we used the definition of the conditional VF (equation 13) with $\kappa(t) = 1 - t$, in the third equality we used Theorem 3.1 and equation 12, and in the fourth equality we used equation 23. Since $x_t$ realizes a minimum of the length function, it is a geodesic. $\square$

# D ALGORITHMIC COMPARISON TO RIEMANNIAN DIFFUSION MODELS

**Algorithm 2** Riemannian Diffusion Models

**Require:** base distribution $p(x_T)$, target $q(x_0)$
  Initialize parameters $\theta$ of $s_t$
  **while** not converged **do**
    sample time $t \sim \mathcal{U}(0, T)$
    sample training example $x_0 \sim q(x_0)$

    % simulate Geometric Random Walk
    $x_t = \texttt{solve\_SDE}([0, t], x_0)$

    **if** denoising score matching **then**
      % approximate conditional score
      $\nabla \log p_t(x|x_0) \approx \begin{cases} \text{eig-expansion} \\ \text{Varhadan} \end{cases}$
      $\ell(\theta) = \|s_t(x_t; \theta) - \nabla \log p_t(x|x_0)\|_g^2$
    **else if** implicit score matching **then**
      % estimate Riemmanian divergence
      sample $\varepsilon \sim \mathcal{N}(0, I)$
      $\mathrm{div}_g s_t \approx \varepsilon^\mathsf{T} \frac{\partial s_t}{\partial x_t} \varepsilon + \frac{1}{2} s_t^\mathsf{T} \frac{\partial \log \det g(x_t)}{\partial x_t}$
      $\ell(\theta) = \frac{1}{2} \|s_t(x_t; \theta)\|_g^2 + \mathrm{div}_g s_t$
    **end if**

    $\theta = \texttt{optimizer\_step}(\ell(\theta))$
  **end while**

**Algorithm 3** Riemannian Flow Matching

**Require:** base distribution $p(x_0)$, target $q(x_1)$
  Initialize parameters $\theta$ of $v_t$
  **while** not converged **do**
    sample time $t \sim \mathcal{U}(0, 1)$
    sample training example $x_1 \sim q(x_1)$
    sample noise $x_0 \sim p(x_0)$

    **if** simple geometry **then**
      $x_t = \exp_{x_0}(\kappa(t) \log_{x_0}(x_1))$
    **else if** general geometry **then**
      $x_t = \texttt{solve\_ODE}([0, t], x_0, u_t(x|x_1))$
    **end if**

    % closed-form regression target $u_t(x_t|x_1)$
    $\ell(\theta) = \|v_t(x_t; \theta) - u_t(x_t|x_1)\|_g^2$

    $\theta = \texttt{optimizer\_step}(\ell(\theta))$
  **end while**

Figure 7: Algorithmic comparison between Riemannian Diffusion Models (De Bortoli et al., 2022; Huang et al., 2022) and our Riemannian Flow Matching. Note time is reversed between these formulations. In red, we denote expensive computational aspects (sequential simulation during training), biased approximations (for the score function), and stochastic estimation (for divergence) that may not scale well. Also note that Geometric Random Walk does not converge to the stationary prior distribution unless simulated for an infinite amount of time, in practice requiring tuning $T$ as a hyperparameter depending on the manifold. On simple manifolds, Riemannian Flow Matching bypasses all computational inconveniences and in particular is completely simulation-free.

# E LIMITATIONS

As can be seen from Figure 7, our method still requires simulation of $x_t$ on general manifolds. This sequential process can be time consuming, and a more parallel or simulation-free approach to constructing $x_t$ would be more favorable. Furthermore, the spectral distances require eigenfunction solvers which may be computationally expensive on complex manifolds. Using approximate methods such as neural eigenfunctions (Pfau et al., 2018; Deng et al., 2022) may be a possibility. One major advantage of our premetric formulation is that these eigenfunctions need not be perfectly solved in order to satisfy the relatively simple properties of our premetric.

# F ADDITIONAL FIGURES

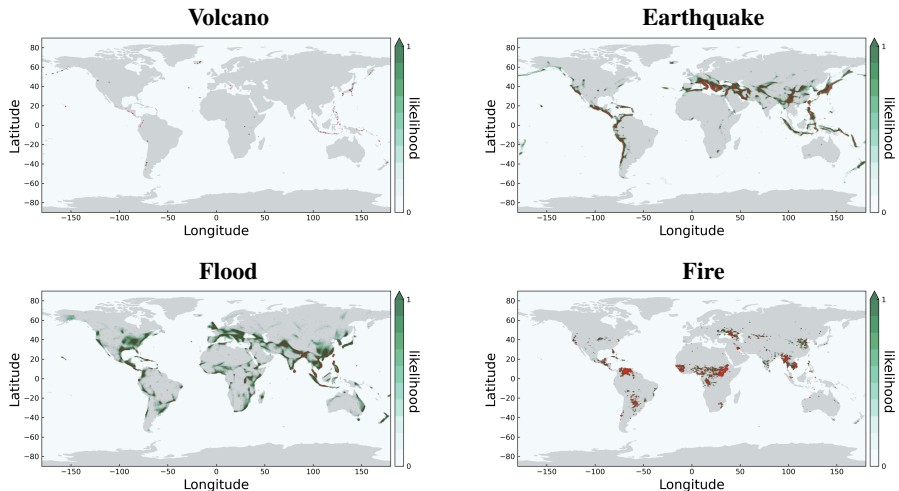

Figure 8: Data samples and Ramachandran plots depicting log likelihood for protein datasets.

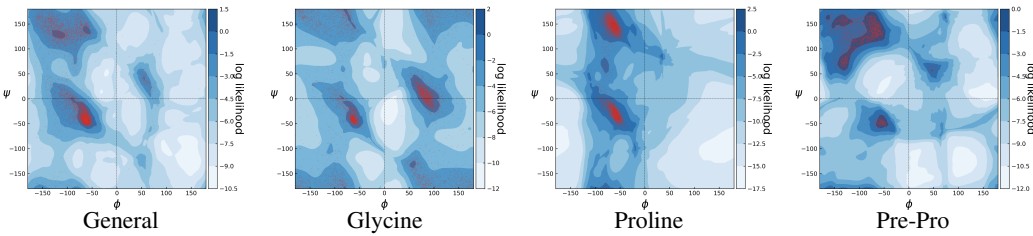

Figure 9: Data samples and Ramachandran plots depicting log likelihood for protein datasets.

# G ADDITIONAL DISCUSSION

## G.1 ON THE USE OF APPROXIMATE SPECTRAL DISTANCES AS THE PREMETRIC

**Computation cost.** The smallest $k$ eigenvalues and their eigenfunctions need only be computed once as a pre-processing step. On manifolds represented as discrete triangular meshes, this step can be done in a matter of seconds. Afterwards, spectral distances can be computed very efficiently for all pairs of points. Note however, that training with RCFM does still require simulating for $x_t$, but as the vector fields (equation 13) do not contain neural networks, the flows can be solved efficiently in practice. This results in a similar cost to diffusion-based methods (Huang et al., 2022; De Bortoli et al., 2022) that require simulation even for simple manifolds.

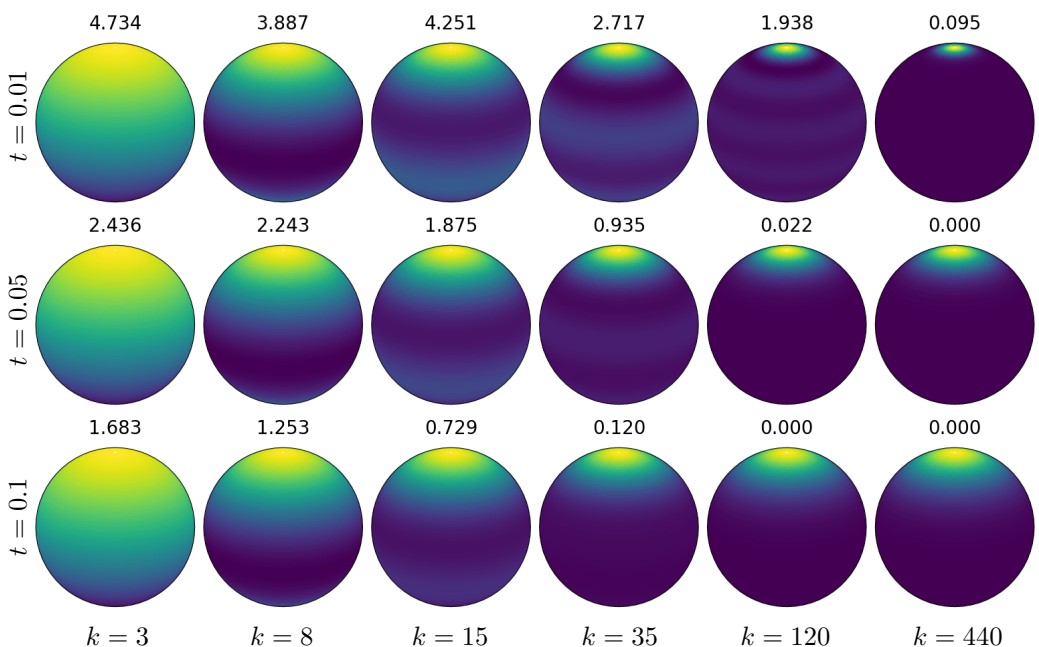

Figure 10: A visualization of the approximated heat kernel in equation 24 on a 2D sphere for different $k$ values. Above each visualization we show the relative error of the conditional score function: $\mathbb{E}_{p_t(x_t|x_0)}\left[\|\nabla_{x_t}\log\tilde{p}_t(x_t|x_0) - \nabla_{x_t}\log p_t(x_t|x_0)\| / \|\nabla_{x_t}\log p_t(x_t|x_0)\|\right]$ where $x_0$ is located at the top of the sphere. We see that especially at small $t$ values (near data distribution), there is a significant error in the score function approximation even when using hundreds of eigenfunctions.

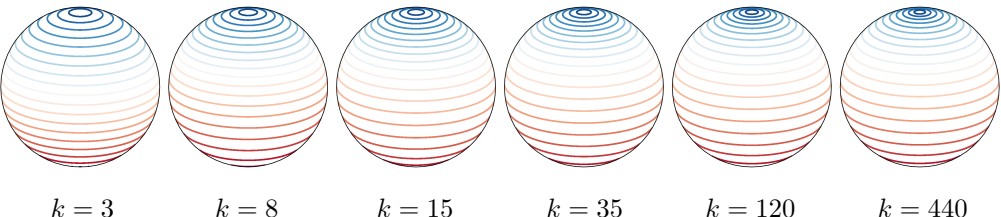

Figure 11: Spectral distances (biharmonic), *i.e.*, $\mathrm{d}(x, \cdot)$ where $x$ is at the top of the sphere, are visualized using isolines. We see that it is very easy to satisfy the properties of a premetric with very few eigenfunctions, importantly the *Positive* property: $\mathrm{d}(x, y) = 0$ iff $x = y$, which allows us to concentrate probability perfectly at every point of the manifold, and the *Non-degenerate* property $\nabla\mathrm{d}(x, y) \neq 0$ iff $x \neq y$ (satisfied here everywhere except at the antipodal point) that allows to properly define the flow in equation 13 almost everywhere.

**Sufficiency with finite $k$.** One may wonder if we pay any approximation costs when using finite $k$; the answer is no. In fact, we only need as many eigenfunctions as it takes to be able to distinguish (almost) every pair of points on the manifold. Put differently, we don't need to compute the spectral distances perfectly, only sufficiently enough that the conditions of our premetric are satisfied. Regarding the question of what $k$ is enough? This is only understood partially: for local neighborhoods the number of required eigenfunctions is the manifold dimension (but not necessarily the first ones), a property proven in (Jones et al., 2008, Theorem 2). Nevertheless, the use of spectral distances computed with $k$ smallest eigenvalues is equivalent to computing Euclidean distances in a $k$-dimensional Euclidean embedding using the same eigenfunctions; this embedding is known to preserve neighborhoods optimally (Belkin & Niyogi, 2003).

As comparison, Riemannian score-based generative models (De Bortoli et al., 2022) suggests a heat kernel approximation

$$\tilde{p}_t(x_t|x_0) = \sum_{i=0}^{k} e^{-\lambda_i t} \varphi_i(x_0)\varphi_i(x_t), \qquad (24)$$

resulting in the approximated conditional score function

$$\nabla_{x_t} \log \tilde{p}_t(x_t|x_0) = \nabla_{x_t} \log \sum_{i=0}^{k} e^{-\lambda_i t} \varphi_i(x_0)\varphi_i(x_t). \qquad (25)$$

However, equation 24 is only correct as $k \to \infty$. This is manifested in practice as large amounts of error even when using hundreds of eigenfunctions. In Figure 10, we visualize the heat kernel approximation in equation 24 as well as relative error in the score function, taken in expectation w.r.t. $p_t(x|x_0)$, *i.e.*, the relative error is weighted higher in regions where we will evaluate the conditional score function during training. We see that at small time values close to the data distribution, the conditional score function may not even have the first significant digit correct (in expectation).

In contrast, we visualize the spectral distances using the biharmonic formulation in Figure 11, where we see that we already have an extremely accurate premetric with very few eigenfunctions. In particular, the required properties of a premetric are satisfied even for just $k = 3$, roughly corresponding to the manifold dimension. Higher values of $k$ simply refine the spectral distance but are not necessary.

### G.2 MANIFOLDS WITH BOUNDARY

In considering general geometries, we also consider the case where $\mathcal{M}$ has a boundary, denoted $\partial\mathcal{M}$. In this case, we need to add another condition to our premetric to make sure $u_t(x|x_1)$ will not flow particles outside the manifold. Let $n(x) \in T_x\mathcal{M}$ denote the interior-pointing normal direction at a boundary point $x \in \partial\mathcal{M}$. We add the following condition to our premetric:

4. *Boundary:* $\langle \nabla \mathrm{d}(x,y), n(x) \rangle_g \leq 0, \forall y \in \mathcal{M}, x \in \partial\mathcal{M}$.

If the premetric satisfies this condition, then the conditional VF in equation 13 satisfies

$$\langle u_t(x|x_1), n(x) \rangle_g \geq 0$$

implying that the conditional vector field does not point outwards on the boundary of the manifold.

**Spectral distances at boundary points.** In case $\mathcal{M}$ has boundary we want to make sure the spectral distances in equation 16 satisfy the boundary condition. To ensure this, we can simply solve eigenfunctions $\varphi_i$ using the natural, or Neumann, boundary conditions, *i.e.*, their normal derivative at boundary points vanish, and we have $\langle \nabla_g \varphi_i(x), n(x) \rangle_g = 0$ for all $x \in \partial\mathcal{M}$. This property implies that $\langle \nabla_x \mathrm{d}_w(x,y)^2, n(x) \rangle_g = 0$, satisfying the boundary condition of the premetric.

## H  EXPERIMENT DETAILS

**Training setup.** All experiments are run on a single NVIDIA V100 GPU with 32GB memory. We tried our best to keep to the same training setup as prior works (Mathieu & Nickel, 2020; De Bortoli et al., 2022; Huang et al., 2022); however, as their exact data splits were not available, we used our own random splits. We followed their procedure and split the data according to 80% train, 10% val, and 10% test. We used seeds values of 0-4 for our five runs. We used the validation set for early stopping based on the validation NLL, and then only computed the test NLL using the checkpoint that achieved the best validation NLL. We used standard multilayer perceptron for parameterizing vector fields where time is concatenated as an input to the neural network. We generally used 512 hidden units and tuned the number of layers for each type of experiment, ranging from 6 to 12 layers. We used the Swish activation function (Ramachandran et al., 2017) with a learnable parameter. We used Adam with a learning rate of 1e-4 and an exponential moving averaging on the weights (Polyak & Juditsky, 1992) with a decay of 0.999 for all of our experiments.

**High-dimensional tori.** We use the same setup as De Bortoli et al. (2022). The data distribution is a wrapped Gaussian on the high-dimensional tori with a uniformly sampled mean and a scale of 0.2.

Table 5: Riemannian manifolds with known geodesics that we use in our experiments. The operator $\oplus$ denotes Möbius addition. Exponential maps, logarithm maps, and inner products are used during training with Riemannian Conditional Flow Matching. Note $x, y \in \mathcal{M}$, $u, v \in T_x\mathcal{M}$, and $\|\cdot\|_g^2 = \langle\cdot, \cdot\rangle_g$. The last column $\log|g(x)|$ denotes the logarithm of the absolute value of the determinant of the metric tensor at $x \in \mathcal{M}$; this is used during log-likelihood computation (see equations 37 and 33).

| Manifold $\mathcal{M}$ | $\exp_x(u)$ | $\log_x(y)$ | $\langle u, v \rangle_g$ | $\log|g(x)|$ |
|---|---|---|---|---|
| $N$-D sphere $\{x \in \mathbb{R}^{N+1} : \|x\|_2 = 1\}$ | $x\cos(\|u\|_2) + \frac{u}{\|u\|_2}\sin(\|u\|_2)$ | $\arccos(\langle x, y \rangle)\frac{P_x(y-x)}{\|P_x(y-x)\|_2}$ | $\langle u, v \rangle$ | $0$ |
| $N$-D flat tori $[0, 2\pi]^N$ | $(x + u) \% (2\pi)$ | $\text{arctan2}(\sin(y-x), \cos(y-x))$ | $\langle u, v \rangle$ | $0$ |
| $N$-D Hyperbolic $\{x \in \mathbb{R}^N : \|x\|_2 < 1\}$ | $x \oplus \left(\tanh\left(\frac{\|u\|_2}{1-\|x\|_2^2}\right)\frac{u}{\|u\|_2}\right)$ | $\left(1 - \|x\|_2^2\right)\tanh^{-1}\left(\|-x\oplus y\|_2\right)\frac{-x\oplus y}{\|-x\oplus y\|_2}$ | $\frac{4}{(1-\|x\|_2^2)^2}\langle u, v \rangle$ | $N\log\frac{2}{1-\|x\|_2^2}$ |
| $N \times N$ SPD matrices | $X^{\frac{1}{2}}\exp\{X^{-\frac{1}{2}}UX^{-\frac{1}{2}}\}X^{\frac{1}{2}}$ | $X^{\frac{1}{2}}\log\{X^{-\frac{1}{2}}YX^{-\frac{1}{2}}\}X^{\frac{1}{2}}$ | $\text{tr}\left(X^{-1}UX^{-1}V\right)$ | $\frac{N(N-1)}{2}\log(2) + (N+1)\log\det X$ |

We use a MLP with 3 hidden layers of size 512 to parameterize $v_t$, train for 50000 iterations with a batch size of 512. We then report the log-likelihood per dimension (in bits) on 20000 newly sampled data points.

**Triangular meshes.** We use the exact open source mesh for Spot the Cow. For the Stanford Bunny, we downsample the mesh to 5000 triangles and work with this downsampled mesh. For constructing target distributions, we compute the eigenfunctions on a 3-times upsampled version of the mesh, threshold the $k$-th eigenfunction at zero, then normalized to construct the target distribution. This target distribution is uniform on each upsampled triangle, and further weighted by the area of each triangle. On the actual mesh we work with, this creates complex non-uniform distributions on each triangle. We also normalize the mesh so that points always lie in the range of (-1, 1).

**Maze manifolds.** We represent each maze manifold using triangular meshes in 2D. Each cell is represented using a mesh of 8x8 squares, with each square represented as two triangles. If two neighboring cells are connected, then we connect them using either 2x8 or 8x2 squares; if two neighboring cells are not connected (*i.e.*, there is a wall), then we simply do not connect their meshes, resulting in boundaries on the manifold. This produces manifolds represented by triangular meshes where all the triangles are the same size. We randomly create maze structures based on a breadth-first-search algorithm, represent these using meshes, and we then normalize the mesh so that points lie in the range of (0, 1).

**Vector field parameterization.** We parameterize vector fields as neural networks in the ambient space and project onto the tangent space at every $x$. That is, similarly to (Rozen et al., 2021) we model

$$v_t(x) = g(x)^{-\frac{1}{2}}P_{\pi(x)}v_\theta(t, \pi(x)) \tag{26}$$

where $\pi$ is the projection operator onto the manifold, *i.e.*,

$$\pi(x) = \arg\min_{y\in\mathcal{M}}\|x - y\|_g, \tag{27}$$

and $P_y$ is the orthogonal projection onto the tangent space at $y$.

We also normalize the vector field using $g(x)^{-\frac{1}{2}}$, which cancels out the effect of $g$ on the Riemannian norm and makes standard neural network parameterization more robust to changes in the metric, *i.e.*,

$$\left\|g^{-\frac{1}{2}}v\right\|_g^2 = (g^{-\frac{1}{2}}v)^\mathsf{T}g(g^{-\frac{1}{2}}v) = v^\mathsf{T}v = \|v\|_2^2. \tag{28}$$

We found this bypasses the need to construct manifold-specific initialization schemes for our neural networks and leads to more stable training.

**Log-likelihood computation.** We solve for the log-density $\log p_1(x)$, for an arbitrary test sample $x \in \mathcal{M}$, by using the instantaneous change of variables (Chen et al., 2018), namely solve the ODE

$$\frac{d}{dt}\begin{pmatrix} x_t \\ f_t(x) \end{pmatrix} = \begin{pmatrix} v_t(x_t) \\ -\text{div}_g(v_t)(x_t) \end{pmatrix}, \tag{29}$$

where $\text{div}_g$ is the Riemannian divergence. We solve backwards in time from $t = 1$ to time $t = 0$ with the initial conditions

$$\begin{pmatrix} x_1 \\ f_1(x) \end{pmatrix} = \begin{pmatrix} x \\ 0 \end{pmatrix} \tag{30}$$

and compute the desired log-density at $x$ via

$$\log p_1(x) = \log p_0(x_0) - f_0(x). \tag{31}$$

For manifolds that are embedded in an ambient Euclidean space (*e.g.*, hypersphere, flat tori, triangular meshes), the parameterization in equation 26 allows us to compute the Riemannian divergence directly in the ambient space (Rozen et al., 2021, Lemma 2). That is,

$$\operatorname{div}_g(v_t) = \operatorname{div}_E(v_t) = \sum_i \frac{\partial v_t(x)_i}{\partial x_i}. \tag{32}$$

For general manifolds with metric tensor $g$ (*e.g.*, Poincaré ball model of hyperbolic manifold, the manifold of symmetric positive definite matrices), we compute the Riemannian divergence as

$$\operatorname{div}_g(v_t) = \operatorname{div}_E(v_t) + \frac{1}{2} v_t^{\mathsf{T}} \nabla_E \log \det g \tag{33}$$

where $\operatorname{div}_E$ is the standard Euclidean divergence and $\nabla_E = (\frac{\partial}{\partial x_1}, \ldots, \frac{\partial}{\partial x_d})^T$ is the Euclidean gradient.

**Wrapped distributions.** An effective way to define a simple distribution $p \in \mathcal{P}$ over a manifold $\mathcal{M}$ of dimension $d$ is pushing some simple prior $\tilde{p}$ defined on some euclidean space $\mathbb{R}^d$ via a chart $\phi : \mathbb{R}^d \to \mathcal{M}$; for example $\phi = \exp_x : T_x\mathcal{M} \to \mathcal{M}$ the Riemannian exponential map. Generating a sample $x \sim p(x)$ is done by drawing a sample $z \sim \tilde{p}(z)$, $z \in \mathbb{R}^d$, and computing $x = \phi(z)$. To compute the probability density $p(x)$ at some point $x \in \mathcal{M}$, we integrate over some arbitrary domain $\Omega \subset \mathcal{M}$,

$$\int_\Omega p(x) d\operatorname{vol}_x = \int_{\phi^{-1}(\Omega)} p(\phi(z)) \sqrt{\det g(z)} dz, \tag{34}$$

where $g_{ij}(z) = \langle \partial_i \phi(z), \partial_j \phi(z) \rangle_g$, $i, j \in [d]$, is the Riemannian metric tensor in local coordinates, and $\partial_i \phi(z) = \frac{\partial \phi(z)}{\partial z_i}$. From this integral we get that

$$\tilde{p}(z) = p(\phi(z)) \sqrt{\det g(z)} \tag{35}$$

and therefore

$$p(x) = \frac{\tilde{p}(\phi^{-1}(x))}{\sqrt{\det g(\phi^{-1}(x))}}. \tag{36}$$

and in log space

$$\log p(x) = \log \tilde{p}(\phi^{-1}(x)) - \frac{1}{2} \log \det g(\phi^{-1}(x)). \tag{37}$$

**Numerical accuracy.** On the hypersphere, NLL values were computed using an adaptive step size ODE solver (`dopri5`) with tolerances of `1e-7`. On the high dimensional flat torus and SPD manifolds, we use the same solver but with tolerances of `1e-5`. We always check that the solution does not leave the manifold by ensuring the difference between the solution and its projection onto the manifold is numerically negligible.

On general geometries represented using discrete triangular meshes, we used 1000 Euler steps with a projection after every step for evaluation (NLL computation and sampling after training). During training on general geometries, we solve for the path $x_t$ using 300 Euler steps with projection after every step. In order to avoid division by zero during the computation of the conditional vector field in equation 13, we solve $x_t$ from $t = 0$ to $t = 1 - \varepsilon$, where $\varepsilon$ is taken to be 1e-5; this effectively flows the base distribution to a non-degenerate distribution around $x_1$ that approximates the Dirac distribution, similar to the role of $\sigma_{\min}$ of Lipman et al. (2023).

See Hairer et al. (2006) and Hairer (2011) for overviews on ODE solving on manifolds.

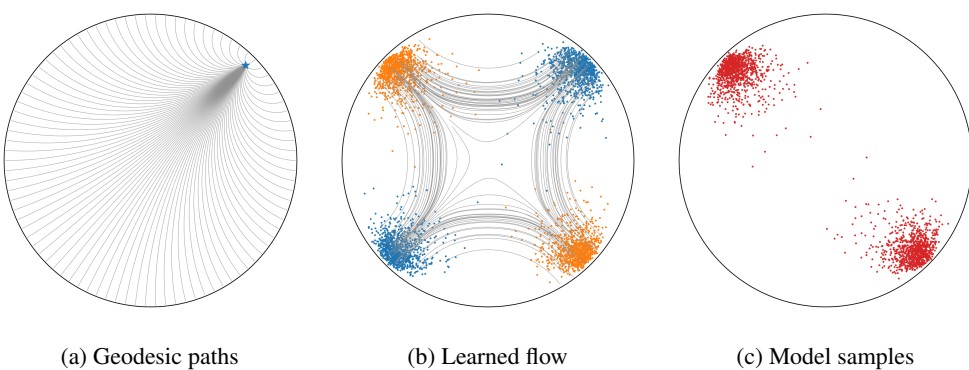

(a) Geodesic paths        (b) Learned flow        (c) Model samples

Figure 12: (a) Geodesic paths on a hyperbolic manifold (represented using the Poincaré disk model) originating from a single point (blue star). (b) Learned CNF where blue samples are from $p(x_0)$ and orange samples are from $q(x_1)$. The CNF respects the geometry of the hyperbolic manifold, and learns to transport along geodesic paths.

## I   Empirical Runtime Estimates

During Riemannian Flow Matching training, there are two main computational considerations: (i) solving for $x_t$, and (ii) computing the training objective. In comparison to diffusion models, Conditional Flow Matching has the clear advantage in (ii), since we don't need to estimate a conditional score function through an infinite series (as in DSM), nor do we require divergence estimation (as in ISM). In the following, we focus on runtime for different ways of (i) solving for $x_t$:

Simulation of ODE/SDE (200 steps) on a *flat torus*:       6.36 iterations / second

**Simulation-free** on a *flat torus*:       104.04 iterations / second

Simulation of ODE/SDE (200 steps) on the *bunny mesh*:       0.422 iterations / second

These numbers were benchmarked on a Tesla V100 GPU, with batch size 64. The runtime is for the full training loop, but the main difference between the three lines is how $x_t$ is solved while all others (i.e. architecture) are fixed.

Generally, the bunny and other mesh manifolds are more expensive due to the projection operator applied after every step, which we implemented rather naively for meshes. However, comparing to iteratively solving an ODE/SDE even on simple manifolds, we see a significant speedup of roughly **17x** even when taking into account the full training loop (including gradient descent etc). This shows the efficiency gains from using simulation-free training over simulation-based.

## J   Additional Experiments

Here we consider manifolds with constrained domains and non-trivial metric tensors, specifically, a hyperbolic space and a manifold of symmetric positive definite matrices, equipped with their standard Riemannian metrics. See Table 5 for a summary of the geometries of these manifolds.

### J.1   Hyperbolic Manifold

We use the Poincaré disk model for representing a hyperbolic space in 2-D. Figure 12 visualizes geodesic paths originating from a single point on the manifold, a learned CNF using Riemannian Conditional Flow Matching, and samples from the learned CNF. Our learned CNF respects the geometry of the manifold and transports samples along geodesic paths, recovering a near-optimal transport map in line with the Riemannian metric. Similarly, due to the use of this metric, the CNF never transports outside of the manifold.

Table 6: Test set evaluation on EEG datasets. Stdev. estimated over 3 runs.

| | | BCI-IV-2b 6×6 (21D) | | BCI-IV-2a 25×25 (325D) | | BCI-IV-1 59×59 (1770D) | |
|---|---|---|---|---|---|---|---|
| Geodesic | Norm | NLL | Valid | NLL | Valid | NLL | Valid |
| Euclidean | Euclidean | -61.58$_{\pm 0.26}$ | **100**$_{\pm 0.00}$ | -276.07$_{\pm 0.66}$ | 81.23$_{\pm 5.12}$ | N/A | 0$_{\pm 0.00}$ |
| Riemannian | Euclidean | -61.64$_{\pm 0.22}$ | **100**$_{\pm 0.00}$ | **-277.06**$_{\pm 0.87}$ | 91.47$_{\pm 5.91}$ | N/A | 0$_{\pm 0.00}$ |
| Euclidean | Riemannian | -52.22$_{\pm 9.67}$ | **100**$_{\pm 0.00}$ | -267.42$_{\pm 5.87}$ | **100**$_{\pm 0.00}$ | -1167.63$_{\pm 40.53}$ | **100**$_{\pm 0.00}$ |
| Riemannian | Riemannian | **-61.76**$_{\pm 0.24}$ | **100**$_{\pm 0.00}$ | -271.54$_{\pm 1.17}$ | **100**$_{\pm 0.00}$ | **-1209.88**$_{\pm 53.55}$ | **100**$_{\pm 0.00}$ |

## J.2 MANIFOLD OF SYMMETRIC POSITIVE MATRICES

We use the space of symmetric positive definite (SPD) matrices with the Riemannian metric (Moakher & Batchelor, 2006). We construct datasets using electroencephalography (EEG) data collected by Blankertz et al. (2007); Brunner et al. (2008); Leeb et al. (2008) for a Brain-Computer Interface (BCI) competition. We then computed the covariance matrices of these signals, following standard preprocessing procedure for analyzing EEG signals (Barachant et al., 2013).

In Table 6, we report estimates of negative log-likelihood (NLL) and the percentage of simulated samples that are valid SPD matrices (*i.e.*, samples which lie on the manifold). We ablate and note the importance of the Riemannian geodesic and the Riemannian norm during training.

**Riemannian geodesic.** We compare using Riemannian geodesics (*i.e.*, setting the premetric to be the geodesic distance) and Euclidean geodesics (*i.e.*, setting the premetric to be the $L_2$ distance). In comparing between different paths, we find that the Riemannian one generally performs better as it respects the geometry of the underlying manifold. Figure 13 visualizes the space of $2 \times 2$ SPD matrices as a convex cone. It displays how the Riemannian geodesic behaves—its flow always becomes perpendicular to the boundary when it gets close to boundary and therefore does not leave the manifold—whereas the Euclidean geodesic ignores this geometry.

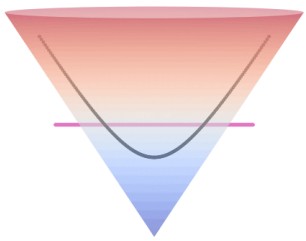

Figure 13: Comparison of extrapolated geodesic paths on SPD manifold, visualized as a convex cone.
(*black*) Riemannian geodesic.
(*violet*) Euclidean geodesic.

**Riemannian norm.** We also compare between using the Riemannian norm (*i.e.*, $\|\cdot\|_g^2$ for training and using the Euclidean norm (*i.e.*, $\|\cdot\|_2^2$). Theoretically, the choice of norm does not affect the optimal $v_t(x)$, which will equal to $u_t(x)$; however, when $v_t$ is modeled with a limited capacity neural network, the choice of norm can be very important as it affects which regions the optimization focuses on (in particular, regions with a large metric tensor). In particular, on the SPD manifold, similar to hyperbolic, the metric tensor increases to infinity in regions where the matrix is close to being singular (*i.e.*, ill-conditioned). We find that especially for larger SPD matrices, using the Riemannian norm is important to ensure $v_t$ does not leave the manifold during simulation (that is, the simulated result is still a SPD matrix).

