# OpenReview forum: "Flow Matching on General Geometries"
_ICLR.cc/2024/Conference — ICLR 2024 oral_

### Official Review · Reviewer_q47k · 2023-10-24

**Soundness:** 4 excellent
**Presentation:** 4 excellent
**Contribution:** 3 good
**Rating:** 8
**Confidence:** 3

**Summary:**

The authors propose a method Riemannian Flow Matching for finding continuous normalizing flows on manifolds. The main idea is to construct flows from noise samples towards samples from the training distribution in such a way that the entire flow concentrates around the specific training example. The flows can be constructed by flowing along gradients of the Riemannian distance, or, as the authors show, gradients of a premetric. Such premetrics can be constructed to give flows that are efficient to evaluate.

**Strengths:**

- well-written and clearly presented paper
- I believe the main idea with the conditional flows arising from premetrics is both interesting and efficient
- the method avoids some of limitations of other approaches including being simulation-free in some cases and avoiding divergence computations

**Weaknesses:**

I have not identified substantial weaknesses.

**Questions:**

No questions.

---

> ### Author Response · Authors · 2023-11-22
> **Authors response**
>
> We thank the reviewer for their commendations and concise review.

---

### Official Review · Reviewer_tLmW · 2023-10-30

**Soundness:** 3 good
**Presentation:** 3 good
**Contribution:** 3 good
**Rating:** 8
**Confidence:** 4

**Summary:**

This work presents a Flow matching framework for generative modeling on Riemannian manifolds, where the paper proposes a novel and simple conditional flow defined on general manifolds through the premetric. Experimental results show that the proposed approach can effectively model data on diverse manifolds and can scale to higher dimensions which previous diffusion models failed.

**Strengths:**

- The paper is well-written and easy to follow with sufficient background and related works.

- The proposed framework provides important advantages over previous methods: (1) does not require divergence computation for training and is thereby scalable to higher dimensions and (2) readily adaptable to general manifolds.

- Especially, the construction of the conditional flows using the premetrics is novel which makes the proposed method applicable to general geometries.

- This work presents the application to data on general manifolds such as manifolds with non-trivial curvature (triangular mesh) or manifolds with boundaries (maze), whereas previous diffusion models were limited to simple manifolds (e.g., sphere, torus) and their product manifolds.

**Weaknesses:**

- The advantage of simulation-free training on simple geometries is not clear. How fast is the training of the proposed approach compared to the training of previous diffusion models? As RSGM (Riemannian Score-based Generative Model) observes that they achieve similar test NLL with only 5 in-training simulation steps (Section O.1), I think that the simulation-free training does not provide significant time efficiency.

- The reason for the outperformance of the proposed framework is not clear. Why does the proposed method outperform the Riemannian Diffusion model? Further, why does CNF Matching show superior performance on Earthquake and Flood datasets over the proposed method?

I would like to raise my score if the above concerns are sufficiently addressed.

**Questions:**

- Please address the questions in the Weakness.

- Why is the result of the proposed method for Flood dataset in Table 2 highlighted in bold even though CNF Matching shows better result?

- Can the other diffusion models applied to general manifolds such as the triangular mesh? For example, RSGM seems to be applicable when using the denoised score matching with the Varadhan approximation.

---

> ### Author Response · Authors · 2023-11-22
> **Authors response**
>
> We thank the reviewer for their commendations and meticulous comments. We address them below.
>
> > How fast is the training of the proposed approach compared to the training of previous diffusion models? As RSGM (Riemannian Score-based Generative Model) observes that they achieve similar test NLL with only 5 in-training simulation steps (Section O.1), I think that the simulation-free training does not provide significant time efficiency.
>
> Note that in Appendix I, we provide wallclock time comparison between simulating the path and our simulation-free approach. We see a 17x improvement in training speed even after taking into account neural network evaluation and gradient descent. This is compared to a 200 step baseline.
>
> While it may be true that fewer steps are needed for some manifolds, we note that this can be a tricky hyperparameter to tune. To the best of our knowledge, in the existing Riemannian diffusion model works [1, 2], the number of steps is tuned on each manifold and can be very different depending on the manifold (ranging from 100 to 1000). On the other hand, being able to perform exact sampling with a simulation-free approach completely foregoes the need to consider such a hyperparameter. Also note that the likelihood values shown in O.1 in the RSGM paper [2] (with up to 200 steps) are still quite inferior to the final results reported in Table 4 (using 1000 steps) in the main part of the paper [2].
>
> > Why does the proposed method outperform the Riemannian Diffusion model?
>
> A potential reason could be that we have zero numerical & approximation errors. Please see Appendix D, where we highlighted, in red, sources of potential errors during training between our framework and Riemannian diffusion. Namely, for diffusion models, both the sample $x_t$ and the “supervision signal” (either denoising or implicit score matching) must be approximated. This is not the case for RFM, where we have both in closed form for simple manifolds. Also, the noising processes in diffusion models do not reach the uniform distribution until infinite time, requiring yet another hyperparameter to tune which is the time horizon of the simulation.
>
> > Further, why does CNF Matching show superior performance on Earthquake and Flood datasets over the proposed method?
>
> Whereas RFM prescribes both the marginal probability path and the marginal vector field (which is being regressed onto), there is no unique vector field for a chosen probability path. CNF Matching takes advantage of this flexibility and prescribes only the marginal probability path and gives the model the freedom to choose any vector field that matches this probability path. It may be this extra freedom that is allowing CNF Matching to perform better. Note however, that CNF Matching has a biased loss that cannot scale as well as RFM.
>
> > Why is the result of the proposed method for Flood dataset in Table 2 highlighted in bold even though CNF Matching shows better result?
>
> Apologies, this is a mistake on our part. It was not our intention to mislead, and we have fixed this. Thanks for catching this. We have also weakened some of our statements surrounding the experiments to reflect that we only achieve state-of-the-art on some but not all data sets that we consider.
>
> > Can the other diffusion models applied to general manifolds such as the triangular mesh? For example, RSGM seems to be applicable when using the denoised score matching with the Varadhan approximation.
>
> There are two approaches for approximating the conditional score function that RSGM [2] proposed: (1) based on a truncated heat kernel expansion using eigendecomposition, and (2) based on the Varadhan approximation. The second option (2) actually will not work on general manifolds as it (i) requires the logarithm map globally and (ii) is only justified for small t values.
>
> For approach (1), the truncated heat kernel expansion using eigendecomposition is possible, but the number of eigenfunctions required for reasonable approximation of the score is very high for RSGM. It is much easier to satisfy the properties of a premetric in our RFM framework---possible with a small finite number—than it is to approximate the heat kernel using eigenfunctions. Indeed, we **expanded the discussion on k and added extra illustrative examples in Appendix G.1** on comparing the number of eigenfunctions required by RSGM and RFM, where we see very large score approximation errors even on a simple 2D-sphere using 100s of eigenfunctions, while spectral distance already provides a valid premetric using 3 eigenfunctions. We believe this property is further exacerbated on more general complex manifolds.
>
> We hope this answers the reviewer’s concerns. If we missed anything, please feel free to let us know.
>
> [1] Riemannian Diffusion Models. https://arxiv.org/abs/2208.07949.
>
> [2] Riemannian Score-Based Generative Modelling. https://arxiv.org/abs/2202.02763.

---

> > ### Comment · Reviewer_tLmW · 2023-11-23
> >
> > Thank you for the detailed response. I have raised my score from 6 to 8.

---

### Official Review · Reviewer_FTs2 · 2023-10-31

**Soundness:** 4 excellent
**Presentation:** 4 excellent
**Contribution:** 3 good
**Rating:** 8
**Confidence:** 4

**Summary:**

This paper proposes Riemannian flow matching (RFM), a generalization of Euclidean flow matching techniques to arbitrary (complete and connected) Riemannian manifolds. The basic idea is similar to the Euclidean setting, where the authors seek to learn a vector field pushing a tractable base distribution to the data distribution, which is achieved by conditional flow matching on a specified path of probability measures. However, the Riemannian setting introduces several challenges, chiefly in the construction of said probability paths and corresponding flows. The authors demonstrate how to construct such flows using general premetrics, a special case of which is the geodesic distance. Practically, the method enables is simple, scalable approach for learning continuous-time normalizing flows on general manifolds.

**Strengths:**

- The problem of learning models with geometric constraints is both difficult and important, and this paper solves several key practical challenges present in prior work in this area (namely, simulation-free and scalable training).
- The paper is a joy to read -- the paper is atypically clear in its aims, methods, and results.
- The empirical support for the method is clear and convincing across a diverse set of settings, and the evidence provided by the experiments generally supports the conclusions drawn by the authors.
- All theoretical claims made in the paper are, to the best of my knowledge, complete and correct.

**Weaknesses:**

- It would have been interesting to have an experiment exploring the effect of $k$ on the learned model (i.e. the number of terms taken in the spectral distance, Equation 16).
- In the same vein, additional experiments on the choice of scheduler $\kappa(t)$ could provide valuable practical insights regarding the design of these models.

**Questions:**

- The base measure used throughout the paper is the uniform distribution. Is there any benefit to considering non-uniform base distributions? While the theory easily admits arbitrary base distributions, are there practical challenges in using non-uniform distributions?
- Some more exposition regarding practical choices surrounding premetric choices would improve the paper. For instance, can we say anything about what makes a particular premetric "good"? Are there principled reasons someone would choose the diffusion distance over the biharmonic distance? Are spectral distances chosen in Section 3.2 primarily for practical reasons, or are there other aspects that make the corresponding premetrics desirable? Is there a compelling reason spectral distances use the spectrum of Laplace-Beltrami operator, rather than some other operator?

---

> ### Author Response · Authors · 2023-11-22
> **Authors response**
>
> We thank the reviewer for their commendations and insightful questions. Below we address their main points:
>
> > the effect of k (the number of eigenfunctions) on the learned model.
>
> Theoretically we know there exists a finite (bounded) value of k that allows the spectral distances to satisfy our premetric conditions. To help provide intuition regarding the effect of $k$, we have **expanded the discussion and added a simple experiment in Appendix G.1** where we compare spectral distance and diffusion model’s conditional probability on the 2D-sphere for different choices of $k$. The spectral distance changes only slightly for different $k$ and already separates all points for $k=3$ and is a valid premetric. Alongside this, we compare to diffusion models, which in contrast, require many more eigenfunctions to be able to concentrate probability mass at small time values $t$ and to get an accurate estimate of the conditional score function. In fact, regardless of how many eigenfunctions are used for diffusion the score approximation is always biased.
>
>
> > choice of scheduler.
>
> We believe the optimal choice of scheduler depends on the data distribution itself, and it is difficult to infer at the time of training how to best choose a scheduler. For instance, [1] chooses a scheduler that minimizes the kinetic energy of the learned model (for the Euclidean case). It could be interesting to see how that translates to manifolds, but it does not seem trivial.
>
>
> > Is there any benefit to considering non-uniform base distributions? While the theory easily admits arbitrary base distributions, are there practical challenges in using non-uniform distributions?
>
> We note first that the maze manifolds use Gaussian mixture models as base and target distributions, and the high-dimensional tori experiment uses a wrapped Gaussian. It is actually often harder to use a uniform distribution because in high dimensions, the volume grows exponentially (so density decreases exponentially). For manifolds with boundaries, ideally the base distribution does not have significant mass on the boundary itself. Overall, we aren’t aware of any practical challenges of using non-uniform distributions. However, choosing a good base distribution is also unclear a priori; a good base distribution for generative modeling is perhaps one that is already close to the data distribution.
>
>
> > For instance, can we say anything about what makes a particular premetric "good"?
>
> We agree that being able to answer such questions can bring many insights into this and similar frameworks. However, we don’t think there is a clear cut answer. Similar to the scheduler, it may very well be that a “good” premetric depends on the data distribution.
>
>
> > Are there principled reasons someone would choose the diffusion distance over the biharmonic distance?
>
> Diffusion distances have a time parameter ($\tau$) that can be tweaked to achieve certain properties. For example, as can be seen in Figure 2 in [1], that long time parameter provides a smoother overall distance field, while short time parameters is more isotropic near the source point at the cost of a more wiggly distance landscape. Practically, we note that the biharmonic is often easier to use as a first choice because of the lack of hyperparameters, where diffusion distance requires strict tuning of the time parameter which we found it is quite sensitive to.
>
>
> > Are spectral distances chosen in Section 3.2 primarily for practical reasons, or are there other aspects that make the corresponding premetrics desirable?
>
> Spectral distances have a certain computational benefit over geodesic distances as they can be computed on the fly between any pair of points on the manifold by simply computing the euclidean distance following a weighted eigenfunction embedding of these points. Apart from that, diffusion distances are intuitively constructed by averaging many random walks, so they end up being smoother and more robust to topological noise (e.g., holes, narrow passages). See https://www.pnas.org/doi/abs/10.1073/pnas.0500334102, e.g., Figure 2, which initially motivated diffusion distances and diffusion maps. We see the benefit of spectral distances in Figure 1 (bottom row), where the geodesic (bottom-left) wants to always transport along the boundary, but this is not ideal as the marginal distribution ends up being a Dirac on the boundary. Spectral distances (bottom-right) avoid the boundary and allow for a diffeomorphic flow of mass within the manifold.
>
>
> > Is there a compelling reason spectral distances use the spectrum of Laplace-Beltrami operator, rather than some other operator?
>
> The eigenfunctions of the Laplace-Beltrami (LB) operator are considered to be the most natural (or even, the “correct”) generalization of the Fourier basis on the unit circle (arguably the simplest 1D compact manifold) to general manifolds.
>
> [1] On Kinetic Optimal Probability Paths for Generative Models. https://arxiv.org/abs/2306.06626.

---

> > ### Comment · Reviewer_FTs2 · 2023-11-22
> >
> > Thanks for the detailed response!
> >
> > The new experiment in Appendix G.1 is an excellent addition and adds a lot of insight.
> >
> > I think including some of this discussion to the main paper would strengthen the submission further, but I don't think it's strictly necessary since some of these points are somewhat speculative or answered already in the literature.

---

> > > ### Author Response · Authors · 2023-11-22
> > >
> > > Thank you. We agree, it also brought a lot of insight for us and it shows a significant advantage of the premetric approach. Given the strict page limits this year, we only provided the full discussion in the Appendix for now, but we will think about how to squeeze a brief mention/summary into the main paper. Thank you again for the discussion.

---

### Meta-Review · Area_Chair_3yu1 · 2023-12-07

**Metareview:**

Summary:

This paper proposes Riemannian flow matching (RFM), a generalization of Euclidean flow matching techniques to arbitrary (complete and connected) Riemannian manifolds. The basic idea is similar to the Euclidean setting, where the authors seek to learn a vector field pushing a tractable base distribution to the data distribution, which is achieved by conditional flow matching on a specified path of probability measures. However, the Riemannian setting introduces several challenges, chiefly in the construction of said probability paths and corresponding flows. The authors demonstrate how to construct such flows using general premetrics, a special case of which is the geodesic distance. Practically, the method enables is simple, scalable approach for learning continuous-time normalizing flows on general manifolds.

Strengths:

- Learning models with geometric constraints is difficult and important. This paper solves several key practical challenges present in prior work in this area: simulation-free and scalable training.
- The paper is a joy to read -- the paper is atypically clear in its aims, methods, and results.
- The empirical support for the method is clear and convincing across a diverse set of settings, and the evidence provided by the experiments generally supports the conclusions drawn by the authors.
- All theoretical claims made in the paper are complete and correct.
- The paper is well-written and easy to follow with sufficient background and related works.
- The proposed framework provides important advantages over previous methods: (1) does not require divergence computation for training and is thereby scalable to higher dimensions and (2) readily adaptable to general manifolds.
- The construction of the conditional flows using the premetrics is novel which makes the proposed method applicable to general geometries.
- Previous models were limited to simple manifolds (e.g., sphere, torus) and their product manifolds.
- well-written and clearly presented paper
- the main idea with the conditional flows arising from premetrics is both interesting and efficient
- the method avoids some of limitations of other approaches including being simulation-free in some cases and avoiding divergence computations

Weaknesses:

- interesting to have an experiment exploring the effect of on the learned model (i.e. the number of terms taken in the spectral distance, Equation 16).
- additional experiments on the choice of scheduler could provide valuable practical insights regarding the design of these models.
- The advantage of simulation-free training on simple geometries is not clear.
- the simulation-free training does not provide significant time efficiency.
- The reason for the outperformance of the proposed framework is not clear.

Recommendation:

All reviewers vote for acceptance. I, therefore, recommend acccepting the paper and encourage the authors to use the feedback provided to improve the paper for the camera ready version.

**Justification For Why Not Higher Score:**

N/A

**Justification For Why Not Lower Score:**

All reviewers agree on strong acceptance decisions. Few weaknesses and many significant strengths:

- Learning models with geometric constraints is difficult and important. This paper solves several key practical challenges present in prior work in this area: simulation-free and scalable training.
- The paper is a joy to read -- the paper is atypically clear in its aims, methods, and results.
- The empirical support for the method is clear and convincing across a diverse set of settings, and the evidence provided by the experiments generally supports the conclusions drawn by the authors.
- All theoretical claims made in the paper are complete and correct.
- The proposed framework provides important advantages over previous methods: (1) does not require divergence computation for training and is thereby scalable to higher dimensions and (2) readily adaptable to general manifolds.
- The construction of the conditional flows using the premetrics is novel which makes the proposed method applicable to general geometries.
- Previous models were limited to simple manifolds (e.g., sphere, torus) and their product manifolds.

---

### Decision · Program_Chairs · 2024-01-16

Accept (oral)